# Uncovering Critical Sets of Deep Neural Networks via Sample-Independent Critical Lifting

## Abstract

This paper investigates the sample dependence of critical points for neural networks. We introduce a sample-independent critical lifting operator that associates a parameter of one network with a set of parameters of another, thus defining sample-dependent and sample-independent lifted critical points. We then show by example that previously studied critical embeddings do not capture all sample-independent lifted critical points. Finally, we demonstrate the existence of sample-dependent lifted critical points for sufficiently large sample sizes and prove that saddles appear among them.

## 1 Introduction

Neural networks have achieved remarkable success in a wide range of applications, but the understanding of their performance is still elusive. Theoretical studies are thus made to uncover such mysteries (Sun et al., 2020). One major focus is the analysis of the loss landscape. This line of study is challenging due to the complicated, various kinds of network structure and loss function, and importantly, its dependence on data samples.

Recent research has increasingly focused on how critical points in the loss landscape depend on the training data. A notable direction in this line of work involves the Embedding Principle (Zhang et al., 2022, 2021; Bai et al., 2024), which is motivated by the following question: given the critical points of a neural network, what can be inferred about the critical points of another network, without knowing the specific training samples? Critical embedding operators between neural networks of different widths, such as splitting embeddings, null embeddings, and more general compatible embeddings, have been proposed and studied in Zhang et al. (2022, 2021). Critical lifting operators in depth between networks of varying depths have been proposed and studied in Bai et al. (2024). However, the full extent to which these operators explain sample (in)dependence remains unclear. Parallel to this, many studies have investigated the behavior of critical points when specific information about the samples is known. For instance, Cooper (2021) relates the dimensionality of the global minima manifold to the number of samples in a generic setting, while ref. Zhang et al. (2023) explores a teacher-student setup and reveals a hierarchical, branch-wise structure of the loss landscape near global minima that varies with sample size.

In this paper, we advance the understanding of sample dependence of critical points by focusing on neural networks of different widths that represent the same output function. Our main contributions are as follows:

(a) We introduce a sample-independent critical lifting operator, which maps parameters from a narrower network to a set of parameters in a wider network, preserving both the output function and criticality regardless of the training samples.

(b) We demonstrate that not all sample-independent lifted critical points arise from previously studied embedding operators, thus highlighting a broader structure beyond existing frameworks Zhang et al. (2022, 2021).

(c) We identify a class of output-preserving critical sets that, for sufficiently large sample sizes, generally contain sample-dependent critical points. These sets consist entirely of saddle points for one-hidden-layer networks and contains sample-dependent saddles for multi-layer networks.

## 2 Related Works

**Embedding Principle.** The Embedding Principle (EP) was first observed for two neural networks of different widths, stating that "the loss landscape of any network 'contains' all critical points of all narrower networks" (Zhang et al., 2021). In refs. Zhang et al. (2021, 2022), specific critical embedding operators have been proposed and studied. These are linear operators mapping parameters of a narrower network to a wider one which preserve output function and criticality – the image of a critical point is always a critical point. Earlier works also observe the similar phenomenon for one hidden layer neural networks (Fukumizu and ichi Amari, 2000; Fukumizu et al., 2019). More recently, EP for two neural networks of different depths was observed (Bai et al., 2024). The paper introduces critical lifting operators associating a parameter of a shallower network to a set of parameters of a deeper one, where output function and criticality are preserved. In our work, we use the same idea to define sample-independent critical lifting operators, but we focus on two neural networks of different widths and show that not all sample-independent lifted critical points arise from known embedding operators.

**Sample dependence of critical points.** Attempts have been made to explain how the choice of samples affects the geometry of loss landscape. Many works focus on global minima. In Cooper (2021), it is shown that for generic samples, the global minima is a manifold whose codimension equals the sample size. Ref. Simsek et al. (2021) observes that under the teacher-student setting, part of the global minima of neural networks persist as samples change. In Zhang et al. (2023) this is further emphasized, and it studies how the other (sample-dependent) global minima varies – "gradually vanish" as sample size increases, as well as how it affects the behavior of gradient dynamics nearby. Other works, such as Simsek et al. (2023), study critical points assuming samples have specific distributions. Our work applies to both global and non-global critical points, and we emphasize sample-dependent lifted critical points for sufficiently large sample size, thus complementing the previous studies.

**Analysis of saddles.** It has been shown that gradient dynamics almost always avoid saddles (Lee et al., 2017). Thus, it is essential to discover saddles in loss landscape of neural networks. Refs. Fukumizu and ichi Amari (2000); Fukumizu et al. (2019); Simsek et al. (2021); Zhang et al. (2022, 2021) showed that embedding local minima of a narrower network to a wider one tends to produce saddles. Additionally, research by Venturi et al. and Li et al. revealed that, when the network is heavily overparameterized, saddles not only exist but in fact there are no spurious valleys. Similar patterns have been observed in deep linear networks (Nguyen and Hein, 2017; Nguyen, 2019; Kawaguchi, 2016). In this paper, we show under mild assumptions on the training set-up that for one hidden layer networks, all sample-independent lifted critical points are saddles, and sample-dependent lifted saddles exist for multi-layer networks.

## 3 Preliminaries

Let $\mathbb{N} := \{1, 2, 3, ...\}$. Given $N \in \mathbb{N}$, denote by $\mathbb{R}^N$ the (real) Euclidean space of dimension $N$. Given Lebesgue measurable subsets $E_2 \subseteq E_1 \subseteq \mathbb{R}^N$, *the measure of $E_2$ in $E_1$* refers to the induced Lebesgue measure on $E_1$. For example, we would say $\mathbb{R} \times \{(0, 0)\} \subseteq \mathbb{R}^3$ has zero measure in $\mathbb{R}^2 \times \{0\} \subseteq \mathbb{R}^3$. Then we define our notations and assumptions for neural networks and loss functions as follows.

### 3.1 Fully Connected Neural Networks

For simplicity, we only discuss fully-connected neural networks *without bias terms*. We refer to this network architecture whenever we mention a neural network. An $L$ hidden layer neural network with

parameter size $N$, input dimension $d$ and output dimension $D$ is denoted by $H : \mathbb{R}^N \times \mathbb{R}^d \to \mathbb{R}^D$. It is defined iteratively as follows. First, we define the zero-th layer (input layer) as the identity function, with a redundant parameter $\theta^{(0)}$:

$$H^{(0)}(\theta^{(0)}, x) = x, \quad x \in \mathbb{R}^d.$$

Second, we choose an activation $\sigma : \mathbb{R} \to \mathbb{R}$. Then, for every $l \in \{1, ..., L\}$, let $m_l$ denote the number of neurons at the $l$-th layer. Define the $l$-th layer neurons by

$$H^{(l)}(\theta^{(l)}, x) = [H_{k_l}^{(l)}(\theta^{(l)}, x)]_{k_l=1}^{m_l} = \left[ \sigma \left( w_{k_l}^{(l)} \cdot H^{(l-1)}(\theta^{(l-1)}, x) \right) \right]_{k_l=1}^{m_l},$$

where $m_l$ is the width of $H^{(l)}$, $H_{k_l}^{(l)}$ is the $k_l$-th component of $H^{(l)}$, and $\theta^{(l)} := \left( (w_{k_l}^{(l)})_{k_l=1}^{m_l}, \theta^{(l-1)} \right)$, each $w_{k_l}^{(l)}$ being a vector in $\mathbb{R}^{m_{l-1}}$. Note that with our notation, each $H_{k_l}^{(l)}$ is independent of $w_k^{(l)}$ for all $k \neq k_l$. Finally, define $H(\theta, x) = [a_j \cdot H^{(L)}(\theta^{(L)}, x)]_{j=1}^D$ as the whole neural network, where $\theta := \left( (a_j)_{j=1}^D, \theta^{(L)} \right)$.

**Assumption 3.1.** *Assume that the activation $\sigma : \mathbb{R} \to \mathbb{R}$ is a non-polynomial analytic function.*

This assumption takes into consideration the commonly used activations such as tanh ($\frac{1-e^{-x}}{1+e^{-x}}$), sigmoid ($\frac{1}{1+e^{-x}}$), swish ($\frac{x}{1+e^{-x}}$), Gaussian ($e^{-ax^2}$), etc. Moreover, it is easy to see that when $\sigma$ is analytic, the neurons $\{H^{(l)}\}_{l=1}^L$ are all analytic and thus so is the whole network $H$.

**Definition 3.1** (wider/narrower neural network)**.** *Given two $L$ hidden layer neural networks $H_1, H_2$ both with input dimension $d$, output dimension $D$, and the hidden layer widths $\{m_l\}_{l=1}^L, \{m_l'\}_{l=1}^L$, respectively. We say $H_2$ is a wider network than $H_1$, or $H_1$ a narrower network than $H_2$, if $m_l \leq m_l'$ for all $1 \leq l \leq L$.*

## 3.2 Loss Function

Denote the set of samples as $\{(x_i, y_i)_{i=1}^n\}$, where $(x_i)_{i=1}^n \in \mathbb{R}^{nd}$ are sample inputs and $(y_i)_{i=1}^n \in \mathbb{R}^{nD}$ are sample outputs. Given $\ell : \mathbb{R}^D \times \mathbb{R}^D \to [0, \infty)$, we define the loss function (for neural networks with input dimension $d$ and output dimension $D$) as

$$R(\theta) = \sum_{i=1}^n \ell(H(\theta, x_i), y_i).$$

In this paper, we will often deal with neural networks of different widths. As a slight abuse of notation, we shall use $R$ for the loss function (corresponding to fixed samples $(x_i, y_i)_{i=1}^n$) for all neural networks with the same input and output dimensions. Also note that we shall write $R_S$ when emphasizing the samples $S = \{(x_i, y_i)_{i=1}^n\}$ of $R$.

**Assumption 3.2.** *We consider analytic $\ell$. For each $1 \leq j \leq D$, let $\partial_j \ell$ denote the $j$-th partial derivative for its first entry. We assume that $\ell(p, q) = 0$ if and only if $p = q$, and $\partial_p \ell(p, q) = 0$ if and only if $p = q$. Here $\partial_p \ell(p, q) = [\partial_j \ell(p, q)]_{j=1}^D$ is the gradient of $\ell$ with respect to its first entry.*

**Remark 3.1.** A common example is $\ell(p, q) = |p - q|^2$. In this case, the loss function is the one used in regression: $R(\theta) = \sum_{i=1}^n |H(\theta, x_i) - y_i|^2$.

# 4 Sample Independent and Dependent Lifted Critical Points

**Definition 4.1** (sample-independent critical lifting)**.** *Given two fully-connected neural networks $H_1, H_2$. Denote their parameter spaces by $\Theta_1, \Theta_2$, respectively. For each $\theta_1 \in \Theta_1$ let $\mathcal{S}(\theta_1)$ be the collection of samples for which $\theta_1$ is a critical point:*

$$\mathcal{S}(\theta_1) = \left\{ S = \{(x_i, y_i)_{i=1}^n\} : \nabla R_S(\theta_1) = 0, n \in \mathbb{N} \right\}.$$

*Denote by $\mathcal{C}_{\theta_1, S}$ the set of output and criticality preserving parameters of $H_2$:*

$$\mathcal{C}_{\theta_1, S} = \left\{ \theta_2 \in \Theta_2 : H_2(\theta_2, \cdot) = H_1(\theta_1, \cdot), \nabla R_S(\theta_2) = 0 \right\}.$$

*Define a sample-independent critical lifting operator as a map $\tau$ from $\Theta_1$ to the power set of $\Theta_2$ by*

$$\tau(\theta_1) = \bigcap_{S \in \mathcal{S}(\theta_1)} \mathcal{C}_{\theta_1, S}. \tag{1}$$

**Definition 4.2** (sample-dependent/independent lifted critical points). *Given two fully-connected neural networks $H_1, H_2$. Given $\theta_1$ and $S \in \mathcal{S}(\theta_1)$ as in Definition 4.1. We say a parameter $\theta_2 \in \mathcal{C}_{\theta_1,S}$ is a sample-independent lifted critical point (from $\theta_1$) if $\theta_2 \in \tau(\theta_1) = \bigcap_{S \in \mathcal{S}(\theta_1)} \mathcal{C}_{\theta_1,S}$. Otherwise, we say $\theta_2$ is a sample-dependent lifted critical point.*

**Remark 4.1.** To make the sample-independent critical lifting operator non-trivial we should require that $H_1, H_2$ have the same input and output dimensions – otherwise $\tau(\theta_1) = \emptyset$ for all $\theta_1 \in \Theta_1$. In this work, we further consider the case in which $H_1, H_2$ have the same activation, same depth, but one is wider/narrower than the other.

## 4.1 Sample Independent Lifted Critical Points

Recall that a critical embedding is an affine linear map from the parameter space of a narrower neural network to that of a wider one, which preserves output, representation and criticality (Zhang et al., 2022). In particular, for any samples given, the image of a critical point is always a critical point. So by definition we have the following result summarized from (Zhang et al., 2022, 2021).

**Proposition 4.1.1** (critical embeddings produce sample-independent lifted critical points). *The parameters produced by critical embedding operators are sample-independent lifted critical points.*

In refs. Zhang et al. (2022, 2021) some specific critical embedding operators are proposed and studied – the splitting embedding, null-embedding and general compatible embedding. Unfortunately, these embedding operators are not enough to produce all sample-independent lifted critical points for deep neural networks. This follows from the following example:

**Example.** Consider a three hidden layer neural network with $d$ ($d$ is arbitrary) dimensional input, one dimensional output and hidden layer widths $\{m_1, m_2, m_3\}$:

$$H(\theta, x) = \sum_{k_3=1}^{m_3} a_{1k_3} \sigma \left( \sum_{k_2=1}^{m_2} w^{(3)}_{k_3 k_2} \sigma \left( \sum_{k_1=1}^{m_1} w^{(2)}_{k_2 k_1} \sigma(w^{(1)}_{k_1} \cdot x) \right) \right).$$

Given two such networks $H_1, H_2$ with hidden layer widths $\{m_1, m_2, m_3\}$ and $\{m_1, m_2, m_3 + 1\}$, respectively. Define

$$E_{\text{narr}} = \left\{ \theta_{\text{narr}} = \left( (a_{1k_3})_{k_3=1}^{m_3}, (w^{(3)}_{k_3})_{k_3=1}^{m_3}, 0, 0 \right) \right\},$$

$$E_{\text{wide}} = \left\{ \theta_{\text{wide}} = \left( (a'_{1k_3})_{k_3=1}^{m_3+1}, (w'^{(3)}_{k_3})_{k_3=1}^{m_3+1}, 0, 0 \right) \right\}$$

as subsets in the parameter spaces of $H_1, H_2$, respectively. Then the image of $E_{\text{narr}}$ under the splitting embedding, null-embedding and general compatible embedding (altogether) is a proper subset of $E_{\text{wide}}$. Intuitively, this is because these operators "assign" a relationship between the weights on the added second layer neuron to the parameter in $E_{\text{narr}}$. On the other hand, it is easy to see that all parameters in $E_{\text{narr}}$ and $E_{\text{wide}}$ yield the same, constant zero output function, and are critical points, for *arbitrary* samples $(x_i, y_i)_{i=1}^n$, $n \in \mathbb{N}$. Therefore, the previously studied embedding operators do not produce all sample-independent lifted critical points when mapping $E_{\text{narr}}$ to $E_{\text{wide}}$. In particular, whatever sample we choose, we cannot avoid the sample-independent lifted critical points which are not produced by these embedding operators. See Proposition A.2.1 for details of a proof of the example.

**Remark 4.2.** The example can be generalized to $L \geq 3$ hidden layer neural networks.

## 4.2 Sample Dependent Lifted Critical Points

We now turn our focus to sample-dependent lifted critical points. Starting with the one-hidden-layer, one dimensional output case, we show that under mild assumptions on activation and loss function, sample-dependent lifted critical points are saddles. These results extend to deeper architectures, where we identify a set of output-preserving parameters containing sample-dependent critical point and sample-dependent saddles. For both results, we highlight the requirement on sample size for these critical points to exist.

We start with the one hidden layer, one dimensional output case. For an $m$-neuron-wide one hidden layer neural network, we write it as $H(\theta, x) = \sum_{k=1}^{m} a_k \sigma(w_k \cdot x)$ for simplicity, where $\theta = (a_k, w_k)_{k=1}^m$.

**Proposition 4.2.1** (saddles, one hidden layer). *Given samples $(x_i, y_i)_{i=1}^n$ such that $x_i \neq 0$ for all $i$ and $x_i \pm x_j \neq 0$ for $1 \leq i < j \leq n$. Given integers $m, m'$ such that $m < m'$. For any critical point $\theta_{narr} = (a_k, w_k)_{k=1}^m$ of the loss function corresponding to the samples such that $R(\theta_{narr}) \neq 0$, the set of $(w'_k)_{k=m+1}^{m'} \in \mathbb{R}^{(m'-m)d}$ of weights making the parameter*

$$\theta_{wide} = (a_1, w_1, ..., a_m, w_m, 0, w'_{m+1}, ..., 0, w'_{m'}) \tag{2}$$

*a critical point for the loss function has zero measure in $\mathbb{R}^{(m'-m)d}$. Furthermore, any such critical point is a saddle.*

**Remark 4.3.** Due to symmetry of the network structure, the results hold under permutation of the entries of $\theta_{\text{wide}}$.

*Proof.* We show that for a.e. $w'_{m'} \in \mathbb{R}^d$, the partial derivative $\frac{\partial R}{\partial a'_{m'}}$ is non-zero, thus proving the first part of the result. The key to showing such a critical point must be a saddle is that any $\theta_{\text{wide}}$ of the form (2) preserves output function, namely, we have $H(\theta_{\text{narr}}, x) = H(\theta_{\text{wide}}, x)$ for all $x$. See Proposition A.2.2 for more details. □

Then we show that there are sample-dependent lifted critical points when the sample size is larger than the parameter size of the narrower network.

**Theorem 4.2.1** (sample-dependent lifted critical points, one hidden layer). *Assume that $\ell : \mathbb{R} \times \mathbb{R} \to \mathbb{R}$ satisfies: the range of $\partial_p \ell(p, \cdot)$ contains an open interval around 0. Given integers $m, m' \in \mathbb{N}$ such that $m < m'$. Fix $\theta_{narr} = (a_k, w_k)_{k=1}^m$. When sample size $n > 1 + (d+1)m$, there are sample-dependent lifted critical points $\theta_{wide}$ from $\theta_{narr}$ of the form (2). Furthermore, when $n > 2 + (d+1)m$ there are sample-dependent lifted saddles of the form (2).*

**Remark 4.4.** It is clear that for any even integer $s$, $\ell(x, y) = (p - q)^s$ satisfies the hypothesis on $\ell$. In fact, by Lemma A.1.4, this holds for all $\ell$ such that $\ell(p, q) = \ell(p - q, 0)$. We also show in Lemma A.1.5 that the binary cross-entropy loss of distribution $p$ relative to distribution $q$, given by $\ell(p, q) = q \log p + (1 - q) \log(1 - p)$, satisfies this hypothesis.

*Proof.* Specifically, we prove that for any $(x_i)_{i=1}^n \in \mathbb{R}^{nd}$ with $x_i \neq 0$ for all $i$ and $x_i \pm x_j \neq 0$ for $1 \leq i < j \leq n$, and for a.e. $w' \in \mathbb{R}^d$, there are sample outputs $(y_i)_{i=1}^n, (y'_i)_{i=1}^n$ such that

$$\theta_{\text{wide}} = (a_1, w_1, ..., a_m, w_m, 0, w', ..., 0, w')$$

is a critical point for the loss function corresponding to $(x_i, y'_i)_{i=1}^n$, but not so to $(x_i, y_i)_{i=1}^n$. For $N \geq 2 + (d+1)m$, we can choose $(y'_i)_{i=1}^n$ so that not all $\ell(H(\theta_{\text{wide}}, x_i), y_i)$'s vanish. □

**Remark 4.5.** Note that for one hidden layer neural networks every sample-dependent lifted critical point either achieves zero loss, or is a saddle. For simplicity, assume that the activation function is an even or odd function. Given a critical point $\theta_{\text{narr}} = (a_k, w_k)_{k=1}^m$ with $R(\theta_{\text{narr}}) \neq 0$. Consider any critical point $\theta_{\text{wide}} = (a'_k, w'_k)_{k=1}^{m'}$ representing the same output function as $\theta_{\text{narr}}$. By linear independence of neurons (see Lemma A.1.1), $a'_{\bar{k}} = 0$ whenever $w'_{\bar{k}} \notin \{w_k, -w_k\}_{k=1}^m$. On the other hand, if $w'_{\bar{k}} \in \{w_k, -w_k\}_{k=1}^m$ then $\theta_{\text{wide}}$ is a sample-independent lifted critical point. Therefore, up to permutation of the entries, a sample-independent lifted critical point from $\theta_{\text{narr}}$ takes the form (2), thus by Proposition 4.2.1 it must be a saddle. Similar argument works for activations with no parity.

Now we generalize the results to multi-layer neural networks whose output dimensions are arbitrary.

**Proposition 4.2.2** (saddles, general case). *Given samples $(x_i, y_i)_{i=1}^n$ with $x_i \neq 0$ for all $i$ and $x_i \pm x_j \neq 0$ for $1 \leq i < j \leq n$. Given integers $\{m_l\}_{l=1}^L, \{m'_l\}_{l=1}^L$ such that $m_l < m'_l$ for every $1 \leq l \leq L$. Consider two $L$ hidden layer neural networks with input dimension $d$, hidden layer widths $\{m_l\}_{l=1}^L, \{m'_l\}_{l=1}^L$, and output dimension $D$. Denote their parameters by $\theta_{narr}, \theta_{wide}$, respectively. Let $\theta_{narr}$ be a critical point of the loss function corresponding to the samples $(x_i, y_i)_{i=1}^n$, such that $R(\theta_{narr}) \neq 0$. Denote the following sets:*

$$E = \left\{ \theta_{wide} = ((a'_j)_{j=1}^D, \theta_{wide}^{(L)}) : H(\theta_{wide}, \cdot) = H(\theta_{narr}, \cdot), a'_j = (a_{j1}, ..., a_{jm_L}, 0, ..., 0) \right\};$$

$$E^* = \{\theta_{wide} \in E : \nabla R(\theta_{wide}) = 0\}.$$

*Namely, $E$ is a set of parameters preserving output function, $E^*$ is the set of parameters in $E$ also preserving criticality. Then $E^* \neq E$. Furthermore, $E^*$ contains saddles.*

**Remark 4.6.** When $D = L = 1$, we recover the one hidden layer, one dimensional output case.

*Proof.* The extra neurons at each layer of the wider network allows us to freely choose the corresponding parameters so that we have some output-preserving $\theta_{\mathrm{wide}}$ with $H^{(L-1)}(\theta_{\mathrm{wide}}, x_i) \neq 0$ for all $i$ and $H^{(L-1)}(\theta_{\mathrm{wide}}^{(L-1)}, x_i) \pm H^{(L-1)}(\theta_{\mathrm{wide}}^{(L-1)}, x_j) \neq 0$ for $1 \leq i < j \leq n$. Since

$$\frac{\partial H}{\partial a'_{jm'_L}}(\theta_{\mathrm{wide}}) = \sum_{i=1}^n \partial_j \ell(H(\theta_{\mathrm{wide}}, x_i), y_i) \sigma \left( w'^{(L)}_{m'_L} \cdot H^{(L-1)}(\theta_{\mathrm{wide}}^{(L-1)}, x_i) \right)$$

$$= \sum_{i=1}^n \partial_j \ell(H(\theta_{\mathrm{narr}}, x_i), y_i) \sigma \left( w'^{(L)}_{m'_L} \cdot H^{(L-1)}(\theta_{\mathrm{wide}}^{(L-1)}, x_i) \right).$$

This reduces to the proof of Proposition 4.2.1. See Proposition A.2.4 for more details. $\quad\square$

Similarly, sample-dependent lifted critical points exist for multi-layer neural networks. The proof of the theorem below follows the same idea as that of Theorem 4.2.1.

**Theorem 4.2.2** (sample-dependent lifted critical points, general case). *Assume that $\ell : \mathbb{R}^D \times \mathbb{R}^D \to \mathbb{R}$ satisfies: the range of $\partial_p \ell(p, \cdot)$ contains a neighborhood around $0 \in \mathbb{R}^D$. Consider two $L$ hidden layer neural networks with the same assumptions as in Proposition 4.2.2. Denote their parameters by $\theta_{\mathrm{narr}}, \theta_{\mathrm{wide}}$, respectively. Denote the parameter size of the narrower network by $N$. Fix $\theta_{\mathrm{narr}}$. Then there are sample-dependent lifted critical points when sample size $n \geq \frac{1+N}{D}$. Furthermore, there are sample-dependent lifted saddles when $n \geq \frac{1+D+\sum_{l=2}^L m_l(m'_{l-1} - m_{l-1}) + N}{D}$.*

**Remark 4.7.** When $D = L = 1$, we recover the one hidden layer, one dimensional output case. Also note that commonly seen losses such as $\ell(p, q) = (p - q)^s, p, q \in \mathbb{R}^D$ for any even number $s$ satisfy the hypothesis on $\ell$.

# 5 Illustration

In this section we illustrate our results in Section 4 through a toy example. In the example, a specific critical point of a one neuron tanh network $H((a, w), x) = a\tanh(wx)$ is lifted to a set of parameters of a two neuron tanh network $H((a_1, w_1, a_2, w_2), x) = a_1\tanh(w_1x) + a_2\tanh(w_2x)$, where $a, w, a_k, w_k, x$ are real numbers. Specifically, we fix $\theta_1 = (1, \bar{w})$ with $\bar{w} = 1.0258$, sample size $n = 4$, sample inputs $(x_1, x_2, x_3, x_4) = (1/4, 1, 4, 16)$ and vary $y_i$'s. We use $\ell : \mathbb{R} \times \mathbb{R} \to \mathbb{R}$, $\ell(p, q) = (p - q)^2$. So

$$R(\theta) = \sum_{i=1}^4 (H(\theta, x_i) - y_i)^2.$$

To make $\theta_1$ a critical point, $(y_i)_{i=1}^4$ should solve the linear system

$$\begin{pmatrix} \tanh(\frac{1}{4}\bar{w}) & \tanh(\bar{w}) & \tanh(4\bar{w}) & \tanh(16\bar{w}) \\ \frac{1}{4}\tanh'(\frac{1}{4}\bar{w}) & \tanh'(\bar{w}) & 4\tanh'(4\bar{w}) & 16\tanh'(16\bar{w}) \end{pmatrix} \begin{pmatrix} \tanh(\frac{1}{4}\bar{w}) - y_1 \\ \tanh(\bar{w}) - y_2 \\ \tanh(4\bar{w}) - y_3 \\ \tanh(16\bar{w}) - y_4 \end{pmatrix} = \begin{pmatrix} 0 \\ 0 \end{pmatrix}.$$

Let $\varepsilon_i := \tanh(\bar{w}x_i) - y_i$ for $1 \leq i \leq 4$. Clearly, the solution set for $(\varepsilon_i)_{i=1}^4$ is a two dimensional subspace in $\mathbb{R}^4$, and varying $(y_i)_{i=1}^4$ is equivalent to varying $(\varepsilon_i)_{i=1}^4$. Numerically, an approximate solution curve for $(\varepsilon_i)_{i=1}^4 = (\varepsilon_i(t))_{i=1}^4$ is given by

$$\{(1 - 6.0689t, -0.5835 + 3.5621t, 0.3 - 0.3t, -0.1 - 0.9t) : t \in \mathbb{R}\}.$$

First, we show that the image of $\theta_1$ under splitting embeddings remains critical, and is independent of the samples. Note that the set of points produced by splitting embeddings is the line $E :=$ $\{(\delta, \bar{w}, 1 - \delta, \bar{w}) : \delta \in \mathbb{R}\}$ and the partial derivatives of the loss function satisfy

$$\frac{\partial R}{\partial a_1}(\theta_2) = \frac{\partial R}{\partial a_2}(\theta_2), \quad \frac{1}{a_1}\frac{\partial R}{\partial w_1}(\theta_2) = \frac{1}{a_2}\frac{\partial R}{\partial w_2}(\theta_2), \quad \forall \theta_2 \in E.$$

Since $w_1 = w_2 = \bar{w}$ is fixed over $E$, we illustrate the vector field

$$(a_1, a_2) \mapsto \left( \frac{\partial R}{\partial a_1}(a_1, \bar{w}, a_2, \bar{w}), \frac{1}{a_1}\frac{\partial R}{\partial w_1}(a_1, \bar{w}, a_2, \bar{w}) \right)$$

as $(a_1, a_2)$ varies, for the samples we randomly choose. This is indicated in Figure 1 below. As we can see, the vector field vanishes (approximately) along the line $\{a_1 + a_2 = 1\}$, which implies that $E$ is critical under these samples.

Second, we consider critical points in the set $E' := \{(1, \bar{w}, 0, w) : w \in \mathbb{R}\}$. According to Proposition 4.2.1, the points in $E'$ are saddles. In the experiment, we fix the samples by setting $(\varepsilon_i)_{i=1}^4 = (1, -0.5835, 0.3, -0.1)\}$ and check the loss values for different $(a_2, w_2)$, meanwhile keeping $(a_1, w_1) = (1, \bar{w})$ fixed. For these samples, there are three critical points in $E'$. As illustrated in Figure 2, the loss function takes values greater and less than $R(\theta_1) \approx 1.4405$ near each of them, thus showing that they are all saddles.

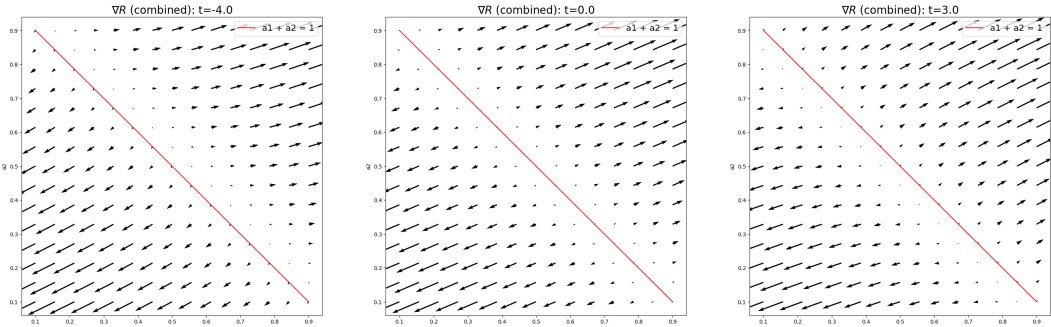

Figure 1: Plot of the vector field $(a_1, a_2) \mapsto \left( \frac{\partial R}{\partial a_1}(a_1, \bar{w}, a_2, \bar{w}), \frac{3}{a_1}\frac{\partial R}{\partial w_1}(a_1, \bar{w}, a_2, \bar{w}) \right)$ for $(a_1, a_2) \in (0.1, 0.9)^2$ with respect to $(\varepsilon_i(-4))_{i=1}^4$ (left), $(\varepsilon_i(0))_{i=1}^4$ (middle) and $(\varepsilon_i(3))_{i=1}^4$. In all three figures, the vector field vanishes approximately along the line $\{a_1 + a_2 = 1\}$, indicating that the parameters produced by splitting embeddings are sample-independent saddles.

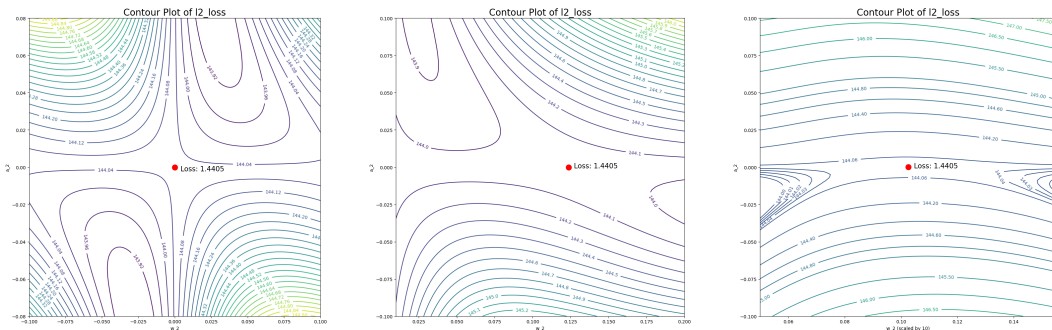

Figure 2: Contour plot of the loss function along the $(w_2, a_2)$-plane with respect to $(\varepsilon_i(0))_{i=1}^4$. The points, marked in red, are approximately $(0, 0)$ (left), $(0.1236, 0)$ (middle) and $(1.0258, 0)$ (right). They correspond to the critical points $(1, \bar{w}, 0, 0), (1, \bar{w}, 0, 0.1236), (1, \bar{w}, 0, 1.0258)$ in $E'$, respectively. From the level curves we can see that these three points are all saddles. Note that in the rightmost figure $w_2$-axis is scaled by 10 for illustration purpose.

Finally, we show the existence of sample-dependent critical points in $E'$. We illustrate this by plotting the zero set of the function

$$(t, w) \mapsto \sum_{i=1}^4 \varepsilon_i(t) \tanh(w x_i).$$

As shown in the proof of Proposition A.2.2, a parameter of the form $(1, \bar{w}, 0, w)$ is a critical point for the loss corresponding to $(\varepsilon_i(t))_{i=1}^4$ if and only if $\varphi(t, w) = 0$. In Figure 3 we can see that for $(t, w) \in (-0.5, 0.5) \times (-0.8, 0.8)$, the zero set of $\varphi$ has two curves; the value of $w$ on the blue curve

varies as $t$ varies, which implies that sample-dependent lifted critical points of the form $(1, \bar{w}, 0, w)$ exist.

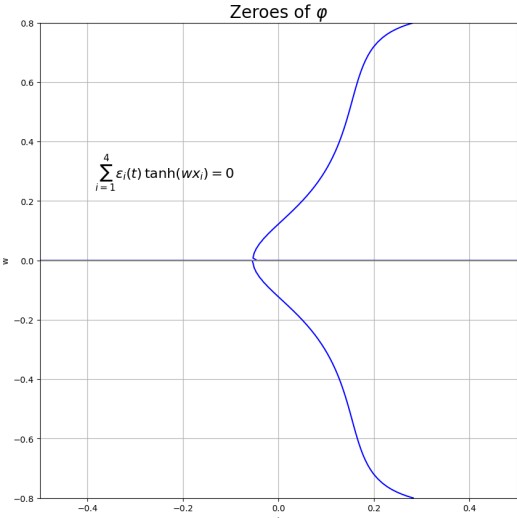

Figure 3: The zero set of $\varphi(t) = \sum_{i=1}^{4} \varepsilon_i(t)\tanh(wx_i)$ for $(t, w) \in (-0.5, 0.5) \times (-0.8, 0.8)$. The blue curve minus the origin, which arises when $t$ ranges approximately from $-0.05$ to $0.3$, is locally the graph of a non-constant function in $t$. This indicates that there is a sample-dependent lifted critical point for each such $t$. Also note that the grey curve $\{(0, t)\}$ indicates a sample-independent lifted critical point $(1, \bar{w}, 0, 0)$. It arises due to the fact that $\tanh(0) = 0$.

# 6    Conclusion and Discussion

In this paper, we propose the sample-independent critical lifting operator (Definition 4.1) and study the sample-independent/dependent lifted critical points. We first show by example that the previously studied critical embeddings may not produce all sample-independent lifted critical points. We then focused on sample-dependent lifted critical points, identifying a specific family of such points and proving that they are necessarily saddles when the loss is non-zero. The sample-independent critical lifting operator provides a way to study the structural aspects of loss landscape dictated purely by the network architecture. Our study of sample-independent critical points reveals the limitation of previously studied embedding operators, suggesting a more delicate relationship between neural networks of different widths. Our study of sample-dependent critical points provides insights into how samples affect the loss landscape.

The paper raises as many questions as the information it provides. First, for sample-independent critical points, we are unclear if all of them are produced by critical embedding operators (not limited to those previously studied ones). We conjecture that they fully characterize all sample-independent lifted critical points for one hidden layer neural networks. Meanwhile, it is interesting to investigate how the completeness of the characterization depends on the network architecture, e.g., choice of activation function, depth/width of network, etc.

Second, we do not have a clear picture about sample-dependent lifted critical points for multi-layer neural networks. Recall that we have shown that all sample-dependent critical points must be of the form (2), but a general form of these points is unclear for multi-layer networks. We expect the existence of additional sample-dependent critical points beyond what we discovered in the paper. Meanwhile, we are interested in the gradient dynamics near the sample-dependent saddles we discovered. Since they are necessarily degenerate and may not have a negative eigenvalue, previous results, e.g., those in Lee et al. (2017) cannot apply immediately.

Third, a better understanding of the sample-independent lifting operator is needed. For example, our construction of sample-dependent lifted critical point requires a specific sample size threshold, which naturally leads to the question whether sample-dependent lifted critical points exist when we keep the sample size fixed while varying samples. More generally, one can study "constrained

sample-independent lifting operator" concerning samples with fixed property. This would help us better understand how different aspects of data affect the loss landscape.

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

## A Appendix

### A.1 Preparing Lemmas

**Lemma A.1.1.** *Let $\sigma : \mathbb{R} \to \mathbb{R}$ be a non-polynomial analytic function. Then for any $d, n \in \mathbb{N}$ and any $x_1, ..., x_n \in \mathbb{R}^d \setminus \{0\}$ with $x_i \pm x_j \neq 0$ for $1 \leq i < j \leq m$, the functions $\{w \mapsto \sigma(w \cdot x_i)\}_{i=1}^n$ are linearly independent.*

*Proof.* We will actually prove a slightly stronger result shown below:

*Let $\sigma : \mathbb{R} \to \mathbb{R}$ be an analytic non-polynomial activation function. Then the following results hold for any $d, m \in \mathbb{N}$ and any $x_1, ..., x_n \in \mathbb{R}^d \setminus \{0\}$*

(a-1) *When $\sigma$ is the sum of a non=zero polynomial and an even/odd analytic non-polynomial, $\{\sigma(w \cdot x_i)\}_{i=1}^n$ are linearly independent if $x_i \pm x_j \neq 0$.*

(a-2) *When $\sigma$ does not have parity and does not satisfy (a-1), then $\{\sigma(w \cdot x_i)\}_{i=1}^n$ are linearly independent if and only if $x_i$'s are distinct.*

(b) *When $\sigma$ is an even or odd function, $\{\sigma(w \cdot x_i)\}_{i=1}^n$ are linearly independent if and only if $x_i \pm x_j \neq 0$ for $1 \leq i < j \leq n$.*

The proof below deals with these cases. For (a-1) we have

- $\sigma$ is the sum of a polynomial and an even, non-polynomial analytic function. Then $\sigma^{(s)}$, the $s$-th derivative of $\sigma$, is an even function for sufficiently large $s$. Since $x_i \pm x_j \neq 0$ for $1 \leq i < j \leq n$, there is some $v \in \mathbb{R}^d$ such that $|x_i \cdot v|$ are distinct and non-zero. It follows from (b) that the (single-variable, even or odd) functions $\{z \mapsto (v \cdot x_i)^s \sigma^{(s)}((v \cdot x_i)z)\}_{i=1}^n$ are linearly independent. Thus, $\{z \mapsto \sigma((v \cdot x_i)z)\}_{i=1}^n$ and thus $\{\sigma(w \cdot x_i)\}_{i=1}^n$ are linearly independent.

- $\sigma$ is the sum of a polynomial and an odd, non-polynomial analytic function. Then $\sigma^{(s)}$ is an odd function for sufficiently large $s$. Argue in the same way as in (a-1) we show the desired result.

For (a-2), note that there are infinitely many even and odd numbers $s_{even}, s_{odd} \in \mathbb{N}$, such that $\sigma^{(s_{even})}(0), \sigma^{(s_{odd})}(0) \neq 0$. Then the result follows from Lemma B.5 in Simsek et al. (2021). One can also refer to other works, such as Zhang et al. (2023).

Then we prove (b). First assume that $\sigma$ is an even function. Then there are even, non-zero numbers $\{s_j\}_{j=1}^\infty$ such that $\sigma^{(s_j)}(0)$, the $s_j$-th derivative of $\sigma$ at 0, is non-zero, for all $j \in \mathbb{N}$. Given $x_1, ..., x_n \in \mathbb{R}^d \setminus \{0\}$ such that $x_i \pm x_j \neq 0$ for $1 \leq i < j \leq n$. Assume $\alpha_1, ..., \alpha_n \in \mathbb{R}$ makes the linear combination of these neurons, $\sum_{i=1}^n \alpha_i \sigma(w \cdot x_i)$, a constant function. Since $x_i \pm x_j \neq 0$ for $1 \leq i < j \leq n$, there is some $v \in \mathbb{R}^d$ such that $|x_i \cdot v|$ are distinct and non-zero. Therefore,

$$z \mapsto \sum_{i=1}^n \alpha_i \sigma\left((v \cdot x_i)z\right) = \text{const.}, \quad \forall z \in \mathbb{R}.$$

Rewriting this in power series expansion near the origin, we obtain

$$\sum_{i=1}^n \alpha_i \sigma\left((v \cdot x_i)z\right) = \sum_{s=0}^\infty \frac{\sigma^{(s)}(0)}{s!} \left(\sum_{i=1}^n \alpha_i (v \cdot x_i)^s\right) z^s = \text{const.}$$

The power series holds for all $z$ in a sufficiently small open interval around 0. Thus, we must have $\sigma^{(s_j)}(0) \sum_{i=1}^n \alpha_i (v \cdot x_i)^{s_j} = 0$ for all $j \in \mathbb{N}$. Let $i_1 \in \{1, ..., n\}$ be (the unique number) such that $|v \cdot x_{i_1}| = \max_{1 \leq i \leq n} |v \cdot x_i|$. If $\alpha_{i_1} \neq 0$ we would have

$$\sum_{i=1}^n \alpha_i (v \cdot x_i)^{s_j} = \Theta\left(v \cdot x_{i_1}\right)^{s_j} \to \infty$$

as $j \to \infty$. Thus, $\alpha_{i_1} = 0$ and we need only consider the rest $n - 1$ neurons. Therefore, by an induction on $n$ we can see that $\alpha_1 = ... = \alpha_n = 0$. This proves the case for even activation.

Then assume that $\sigma$ is an odd function. Again, let $v \in \mathbb{R}^d$ be such that $|v \cdot x_i|$'s are distinct and non-zero. Let $\alpha_1, ..., \alpha_n \in \mathbb{R}$ be such that $\sum_{i=1}^{n} \alpha_i \sigma((v \cdot x_i)z)$ is a constant function in $z$. Its directional derivative along $v$ is given by

$$\frac{\mathrm{d}}{\mathrm{d}z} \left[ \sum_{i=1}^{n} \alpha_i \sigma \left( (v \cdot x_i)z \right) \right] = \sum_{i=1}^{n} \left( \alpha_i (v \cdot x_i) \right) \sigma' \left( (v \cdot x_i)z \right)$$

must also be constant zero. Since $\sigma'$ is an even, analytic, non-polynomial function, our proof above shows that $\alpha_i(v \cdot x_i) = 0$ for all $1 \leq i \leq n$, which then implies $\alpha_i = 0$ for all $1 \leq i \leq n$. Therefore, the neurons are linearly independent.

Conversely, if $x_i - x_j = 0$ for some distinct $i, j$, then we obtain two identical neurons. If $x_i + x_j = 0$ then $\sigma(w \cdot x_i) = \sigma(w \cdot x_j)$ for even function $\sigma$ and $\sigma(w \cdot x_i) + \sigma(w \cdot x_j) = 0$ for odd activation $\sigma$. In either case we obtain two linearly dependent neurons. This completes the proof. $\square$

**Lemma A.1.2.** *Let $N \in \mathbb{N}$ and $g : \mathbb{R}^N \to \mathbb{R}$ a smooth function. Let $x^* \in \mathbb{R}^N$ be a critical point of $g$ such that for any neighborhood $U$ of $x^*$, there is some $x \in U$ with $\nabla g(x) \neq 0$ and $g(x) = g(x^*)$. Then $x^*$ is a saddle.*

*Proof.* We will show that any neighborhood $U$ of $x^*$ contains points $y_1, y_2$ with $g(y_1) < g(x^*) < g(y_2)$. So fix $U$. Choose an $x \in U$ with $\nabla g(x) \neq 0$ and $g(x) = g(x^*)$. Since $\nabla g(x) \neq 0$, the gradient flow $\gamma : [0, \infty) \to \infty$ starting at $x$ is not static; moreover, for some small $\delta > 0$ we have $\gamma[0, \delta) \subseteq U$. Since the value of $g$ is (strictly) decreasing along $\gamma$, we may choose $y_1 := \gamma(\frac{\delta}{2})$, because

$$g \left( \gamma \left( \frac{\delta}{2} \right) \right) < g(\gamma(0)) = g(x) = g(x^*).$$

Similarly, we can find some $y_2 \in U$ with $g(y_2) > g(x^*)$. $\square$

**Definition A.1** ((real) analytic function, rephrase of Defn. 2.2.1 in Krantz and Parks (2002)). *Let $N, M \in \mathbb{N}$ and $\Omega \subseteq \mathbb{R}^N$ be open. A function $f : \Omega \to \mathbb{R}$ is (real) analytic if for each $x \in \Omega$, $f$ can be represented by a convergent multi-variable power series in some neighborhood of $x$. Similarly, a function $f : \Omega \to \mathbb{R}^M$ is (real) analytic if each of its components is real analytic.*

**Remark A.1.** Let $\Omega$ and $U$ be open, and $f, g : \Omega \to \mathbb{R}$, $h : U \to \Omega$ be analytic functions. By Proposition 2.2.2 and Proposition 2.2.8 in Krantz and Parks (2002), $\alpha f + \beta g$, $fg$, $f \circ h$ are analytic functions, i.e., analyticity is preserved by linear combination, multiplication and composition among analytic functions. Moreover, by Proposition 2.2.3 in Krantz and Parks (2002), the partial derivatives of an analytic function are also analytic. In particular, this means when $\sigma$ and $\ell$ are analytic, the neural network, the loss function, and the partial derivatives of the loss function are analytic.

The following lemma is of great importance for the proofs in Section A.2.

**Lemma A.1.3** (Mityagin (2015)). *Let $N \in \mathbb{N}$, $\Omega \subseteq \mathbb{R}^N$ be open and $f : \Omega \to \mathbb{R}$ be analytic. Then either $f$ is constant zero on $\Omega$, or $f^{-1}(0)$ has zero measure in $\Omega$.*

**Lemma A.1.4.** *Let $\ell : \mathbb{R}^2 \to \mathbb{R}$ be a function satisfying Assumption 3.2. Further assume that $\ell(p, q) = \ell(p - q, 0)$ for all $(p, q) \in \mathbb{R}^2$. Then the range of $\partial_p \ell(p, \cdot)$ contains an open interval around 0 for every $p \in \mathbb{R}$.*

*Proof.* Note that we can write $\ell(p, q) = u(p - q)$ for an analytic function $u : \mathbb{R} \to [0, \infty)$, such that $u$ is not constant zero and $u(z) = 0$ if and only if $z = 0$. Since $u$ achieves its minimum at $z = 0$, there is an interval $I$ containing $0 \in \mathbb{R}$ such that $\frac{\mathrm{d}u}{\mathrm{d}z}(z) \geq 0$ for $z \in (0, \infty) \cap I$ and $\frac{\mathrm{d}u}{\mathrm{d}z}(z) \leq 0$ for $z \in (-\infty, 0) \cap I$. Moreover, $z = 0$ is a zero of $\frac{\mathrm{d}u}{\mathrm{d}z}$. Since $u$ is analytic and not constant zero, the zeroes of $\frac{\mathrm{d}u}{\mathrm{d}z}$ is discrete, so by shrinking $I$ if necessary, we would have $\frac{\mathrm{d}u}{\mathrm{d}z}(z) > 0$ for $z \in (0, \infty) \cap I$ and $\frac{\mathrm{d}u}{\mathrm{d}z}(z) < 0$ for $z \in (-\infty) \cap I$. This shows that the range of $\frac{\mathrm{d}u}{\mathrm{d}z}$ contains an open interval around 0.

Now $\partial_p \ell(p, q) = \frac{\mathrm{d}u}{\mathrm{d}z}(p - q)$. Thus,

$$\mathrm{ran}\partial_p \ell(p, \cdot) = \mathrm{ran} \left[ \frac{\mathrm{d}u}{\mathrm{d}z}(p - \cdot) \right] = \mathrm{ran}\frac{\mathrm{d}u}{\mathrm{d}z}.$$

402 It follows that the range of $\partial_p\ell(p,\cdot)$ contains an open interval around 0. □

403 **Lemma A.1.5.** *Let $\ell(p,q) = q\log p + (1-q)\log(1-p)$ for $p, q \in (0,1)$. Then the range of $\partial_p\ell(p,\cdot)$*
404 *contains an open interval around 0 for every $p \in \mathbb{R}$.*

405 *Proof.* This follows from a straightforward computation. Note that $\partial_p\ell(p,q) = \frac{q}{p} - \frac{1-q}{1-p}$ and for each
406 $p$, the derivative of $q \mapsto \partial_p\ell(p,q)$ is a strictly positive constant $\frac{1}{p} + \frac{1}{1-p}$. Since $\partial_p\ell(p,p) = 0$, this
407 implies that for $q$ in a neighborhood $I$ around $p$, $\partial_p\ell(p,I)$ contains an open interval around 0. □

## A.2 Proof of Results

409 **Proposition A.2.1** (Example in Section 4.1). *Assume that $\sigma(0) = 0$. For two three hidden layer*
410 *neural networks, neither the splitting embedding, nor the null embedding operator, nor general*
411 *compatible embedding operator produce all sample-independent lifted critical points.*

412 *Proof.* Let $H$ be a three hidden layer neural network with $d$ ($d \in \mathbb{N}$ is arbitrary) dimensional input,
413 one dimensional output, and hidden width $\{m_1, m_2, m_3\}$. Thus, $H$ can be written as

$$H(\theta, x) = \sum_{k_3=1}^{m_3} a_{1k_3}\sigma\left(\sum_{k_2=1}^{m_2} w_{k_3k_2}^{(3)}\sigma\left(\sum_{k_1=1}^{m_1} w_{k_2k_1}^{(2)}\sigma(w_{k_1}^{(1)} \cdot x)\right)\right).$$

414 Fix arbitrary samples $(x_i, y_i)_{i=1}^n$. Consider parameters for $H$ of the form

$$\theta = \left((a_{1k_3})_{k_3=1}^{m_3}, (w_{k_3}^{(3)})_{k_3=1}^{m_3}, 0, 0\right). \tag{3}$$

415 Namely, all the $w_{k_2}^{(2)}$ and $w_{k_1}^{(1)}$'s are zero vectors. Then, using $\sigma(0) = 0$ we can inductively see that
416 $H^{(1)}(\theta^{(1)}, x) = 0 \in \mathbb{R}^{m_1}$, $H^{(2)}(\theta^{(2)}, x) = 0 \in \mathbb{R}^{m_2}$ and $H^{(3)}(\theta^{(3)}, x) = 0 \in \mathbb{R}^{m_3}$ for all $x$. The
417 partial derivatives for $R$ are as follows. Here $\partial_p\ell$ denotes the partial derivative of $\ell$ with respect to its
418 first entry (note that $\ell : \mathbb{R} \times \mathbb{R} \to \mathbb{R}$).

$$\frac{\partial R}{\partial a_{1\bar{k}_3}} = \sum_{i=1}^n \partial_p\ell(H(\theta, x_i), y_i)H_{\bar{k}_3}^{(3)}(\theta^{(3)}, x_i) = 0.$$

$$\frac{\partial R}{\partial w_{\bar{k}_3\bar{k}_2}^{(3)}} = \sum_{i=1}^n \partial_p\ell(H(\theta, x_i), y_i) \cdot a_{1\bar{k}_3}\sigma'\left(w_{\bar{k}_3} \cdot H^{(2)}(\theta^{(2)}, x_i)\right)H_{\bar{k}_2}^{(2)}(\theta^{(2)}, x_i)$$

$$= \sum_{i=1}^n \partial_p\ell(H(\theta, x_i), y_i) \cdot a_{1\bar{k}_3}\sigma'(0)\sigma(0) = 0.$$

$$\frac{\partial R}{\partial w_{\bar{k}_2\bar{k}_1}^{(2)}} = \sum_{i=1}^n \partial_p\ell(H(\theta, x_i), y_i)$$

$$\cdot \sum_{k_3=1}^{m_3} a_{1k_3}\sigma'\left(w_{k_3}^{(3)} \cdot H^{(2)}(\theta^{(2)}, x_i)\right)w_{k_3\bar{k}_2}^{(3)}\sigma'\left(w_{\bar{k}_2} \cdot H^{(1)}(\theta^{(1)}, x_i)\right)\sigma(w_{\bar{k}_1}^{(1)} \cdot x_i)$$

$$= \sum_{i=1}^n \partial_p\ell(H(\theta, x_i), y_i) \cdot \sum_{k_3=1}^{m_3} a_{1k_3}\sigma'(0)w_{k_3\bar{k}_2}\sigma'(0)\sigma(0) = 0.$$

$$\frac{\partial R}{\partial w_{\bar{k}_1\bar{k}_0}^{(1)}} = \sum_{i=1}^n \partial_p\ell(H(\theta, x_i), y_i)$$

$$\cdot \sum_{k_3=1}^{m_3} a_{1k_3}\sigma'\left(w_{k_3}^{(3)} \cdot H^{(2)}(\theta^{(2)}, x_i)\right)\sum_{k_2=1}^{m_2} w_{k_3k_2}^{(3)}\sigma'\left(w_{k_2}^{(2)} \cdot H^{(1)}(\theta^{(1)}, x)\right)w_{k_2\bar{k}_1}^{(2)}\sigma'(w_{\bar{k}_1} \cdot x_i)(x_i)_{\bar{k}_0}$$

$$= 0 \quad \text{(because } w_{k_2\bar{k}_1}^{(2)} = 0 \text{ for all } k_2\text{)}.$$

419 In other words, we show that any parameter satisfying (3) is a critical point of the loss function,
420 regardless of samples.

Now consider two three hidden layer networks $H, H'$ both with input dimension $d$, output dimension $D$, and hidden layer widths $\{m_l\}_{l=1}^L, \{m_l'\}_{l=1}^L$, respectively. Assume that $m_1' = m_1, m_2' = m_2$, $m_2 > 1$ and $m_3' = m_3 + 1$. In this case, $H'$ is just one neuron wider than $H$ and the embedding of parameters from that of $H$ to $H'$ by general compatible embedding is just splitting embedding or null-embedding. For splitting embedding, note that for any $\theta$ satisfying (3), up to permutation of entries a parameter $\theta'$ given by EP and satisfying (3) takes the form

$$\theta' = \left( (a_{1k_3})_{k_3=1}^{m_3}, (w_1^{(3)}, ..., \delta w_{m_3}^{(3)}, (1-\delta) w_{m_3+1}^{(3)}), 0, 0 \right)$$

for some $\delta \in \mathbb{R}$. In particular, $\delta w_{m_3}^{(3)}, (1-\delta) w_{m_3}^{(3)}$ are parallel vectors in $\mathbb{R}^{m_2}$. However, because $m_2 > 1$, not every $\theta'$ satisfying (3) has two parallel $w_{k_3}^{(3)}$'s. For null embedding, the weight it assigns to the extra neuron is fixed to 0. Thus, these two embedding operators (altogether) do not produce all sample-independent lifted critical points. □

**Remark A.2.** Using the same proof idea, we can show that for two arbitrary $L \geq 3$ hidden layer neural networks, not all sample-independent lifted critical points are produced by these embedding operators.

**Proposition A.2.2** (Proposition 4.2.1 in Section 4.2). *Given samples $(x_i, y_i)_{i=1}^n$ such that $x_i \neq 0$ for all $i$ and $x_i \pm x_j \neq 0$ for $1 \leq i < j \leq n$. Given integers $m, m'$ such that $m < m'$. For any critical point $\theta_{narr} = (a_k, w_k)_{k=1}^m$ of the loss function corresponding to the samples such that $R(\theta_{narr}) \neq 0$, the set of $(w_k')_{k=m+1}^{m'} \in \mathbb{R}^{(m'-m)d}$ of weights making the parameter*

$$\theta_{wide} = (a_1, w_1, ..., a_m, w_m, 0, w_{m+1}', ..., 0, w_{m'}')$$

*a critical point for the loss function has zero measure in $\mathbb{R}^{(m'-m)d}$. Furthermore, any such critical point is a saddle.*

*Proof.* Denote $\theta_{wide} := (a_k', w_k')_{k=1}^m$, so by hypothesis we have $a_k' = 0$ for all $m < k \leq m'$. Note that for any $(w_k')_{k=m+1}^{m'}$, $\theta_{wide}$ preserves output function, i.e., $H(\theta_{wide}, x) = H(\theta_{narr}, x)$ for all $x$. Thus, for any $w_{m'}' \in \mathbb{R}^d$, the partial derivative for $a_{m'}'$ is given by

$$\frac{\partial R}{\partial a_{m'}'}(\theta_{wide}) = \sum_{i=1}^n \partial_p \ell(H(\theta_{wide}, x_i), y_i) \sigma(w_{m'}' \cdot x_i)$$

$$= \sum_{i=1}^n \partial_p \ell(H(\theta_{narr}, x_i), y_i) \sigma(w_{m'}' \cdot x_i).$$

Define

$$\varphi(w_{m'}') = \sum_{i=1}^n \partial_p \ell(H(\theta_{narr}, x_i), y_i) \sigma(w_{m'}' \cdot x_i),$$

so that $\frac{\partial R}{\partial a_{m'}'}(\theta_{wide}) = 0$ if and only if $\varphi(w_{m'}') = 0$. Since i) $\sigma$ is a non-polynomial analytic function, ii) $x_i \neq 0$ for all $i$, and iii) $x_i \pm x_j \neq 0$ for all $1 \leq i < j \leq n$, by Lemma A.1.1 we have that $\{w \mapsto \sigma(w \cdot x_i)\}_{i=1}^n$ are linearly independent. Meanwhile, since $R(\theta_{narr}) \neq 0$, there must be some $i \in \{1, ..., n\}$ with $\ell(H(\theta_{narr}, x_i), y_i) \neq 0$. But then by Assumption 3.2 on $\ell$, we have $H(\theta_{narr}, x_i) \neq y_i$ and thus $\partial_p \ell(H(\theta_{narr}, x_j), y_j) \neq 0$ for some $j \in \{1, ..., n\}$. Therefore, $\varphi$ is a non-trivial linear combination of analytic, linearly independent functions, so it is analytic and not constant zero. But this implies that the set of $\varphi^{-1}(0)$ has zero measure in $\mathbb{R}^d$. It follows that the set of $(w_k')_{k=m+1}^{m'}$ of weights making $\theta_{wide}$ a critical point for the loss function has zero measure in $\mathbb{R}^{(m'-m)d}$.

Let $\theta_{wide}$ be a critical point of the loss function. We now show that it is saddle. Let $U$ be a neighborhood of $\theta_{wide}$. Since $\varphi^{-1}(0)$ has zero measure, $U$ contains a point

$$\theta_{wide}'' = (a_1, w_1, ..., a_m, w_m, 0, w_{m+1}', ..., 0, w_{m'-1}', 0, w_{m'}''),$$

where $w_{m'}'' \notin \varphi^{-1}(0)$, and thus $\nabla R(\theta_{wide}'') \neq 0$. On the other hand, as we mentioned above, $H(\theta_{wide}'', x_i) = H(\theta_{narr}, x_i) = H(\theta_{wide}, x_i)$ for all $i$, whence $R(\theta_{wide}'') = R(\theta_{wide})$. Then Lemma A.1.2 shows that $\theta_{wide}$ is a saddle. □

**Proposition A.2.3** (Theorem 4.2.1 in Section 4.2). *Assume that $\ell : \mathbb{R}^2 \to \mathbb{R}$ satisfies: the range of $\partial_p \ell(p, \cdot)$ contains an open interval around $0 \in \mathbb{R}$. Given integers $m, m', n \in \mathbb{N}$ such that $m < m'$ and $n \geq 1 + (d+1)m$, given $\theta_{narr} = (a_k, w_k)_{k=1}^m$. For any fixed $(x_i)_{i=1}^n \in \mathbb{R}^{nd}$ with $x_i \pm x_j \neq 0$ and for a.e. $w' \in \mathbb{R}^d$, there are sample outputs $(y_i)_{i=1}^n, (y_i')_{i=1}^n$ such that*

$$\theta_{wide} = (a_1, w_1, ..., a_m, w_m, 0, w', ..., 0, w')$$

*is a critical point for the loss function corresponding to $(x_i, y_i')_{i=1}^n$, but not so to $(x_i, y_i)_{i=1}^n$. Furthermore, when $n \geq 2 + (d+1)m$ we can choose $(y_i')_{i=1}^n$ so that $\theta_{wide}$ is a saddle.*

*Proof.* We use the notations in the proof of Proposition A.2.2. Recall that for $\theta_{\text{wide}}$ of the form (2) to be a critical point, we must have $w_{m'}' \in \varphi^{-1}(0)$, where

$$\varphi(w, (y_i)_{i=1}^n) := \varphi(w) = \sum_{i=1}^n \partial_p \ell(H(\theta_{\text{narr}}, x_i), y_i)\sigma(w \cdot x_i).$$

Define

$$M := \begin{pmatrix} | & & | \\ \nabla_\theta H(\theta_{\text{narr}}, x_1) & ... & \nabla_\theta H(\theta_{\text{narr}}, x_n) \\ | & & | \end{pmatrix}.$$

Since $n \geq 1 + (d+1)m$, the kernel of $M$ is non-trivial. Fix $v \in \ker M \setminus \{0\}$. By linear independence of the neurons $\{w \mapsto \sigma(w \cdot x_i)\}_{i=1}^n$, the function $\sum_{i=1}^n v_i \sigma(w \cdot x_i)$ is not constant zero (in $w$), so its zero set has zero measure in $\mathbb{R}^d$ (Lemma A.1.3) and for a.e. $w'$ we have $\sum_{i=1}^n v_i \sigma(w' \cdot x_i) \neq 0$. Then define

$$M' := \begin{pmatrix} | & & | \\ \nabla_\theta H(\theta_{\text{narr}}, x_1) & ... & \nabla_\theta H(\theta_{\text{narr}}, x_n) \\ | & & | \\ \sigma(w' \cdot x_1) & & \sigma(w' \cdot x_n) \end{pmatrix}.$$

and

$$\theta_{\text{wide}} = (a_1, w_1, ..., a_m, w_m, 0, w', ..., 0, w').$$

Notice that for any $k > m$, any $k_0 \in \{1, ..., d\}$, and for any samples $S = \{(x_i, y_i)_{i=1}^n\}$, we have (using $a_k = 0$)

$$\frac{\partial R_S}{\partial w_{k\bar{k}_0}}(\theta_{\text{wide}}) = a_k \cdot \sum_{i=1}^n \partial_p \ell(H(\theta_{\text{narr}}, x_i), y_i)\sigma'(w' \cdot x_i)(x_i)_{\bar{k}_0} = 0.$$

Therefore, $\nabla R_S(\theta_{\text{wide}}) = 0$ if and only if $[\partial_p \ell(H(\theta_{\text{narr}}, x_i), y_i)]_{i=1}^n \in \ker M'$. By our construction above, $v \in \ker M \setminus \ker M'$. Let $v' \in \ker M'$. The hypothesis on $\ell$ implies that the range of the map

$$(q_i)_{i=1}^n \mapsto [\partial_p \ell(H(\theta_{\text{narr}}, x_i), q_i)]_{i=1}^n$$

contains a product neighborhood of $0 \in \mathbb{R}^n$. This implies the existence of $(y_i)_{i=1}^n$ and $(y_i')_{i=1}^n$ such that $[\partial_p \ell(H(\theta_{\text{narr}}, x_i), y_i)]_{i=1}^n$ is a non-zero multiple of $v$ and $[\partial_p \ell(H(\theta_{\text{narr}}, x_i), y_i')]_{i=1}^n$ is a non-zero multiple of $v'$. Then

$$M'[\partial_p \ell(H(\theta_{\text{narr}}, x_i), y_i')]_{i=1}^n = 0, \quad M'[\partial_p \ell(H(\theta_{\text{narr}}, x_i), y_i)]_{i=1}^n \neq 0.$$

In particular, $\varphi(w', (y_i)_{i=1}^n) \neq 0$. Therefore, $\theta_{\text{wide}}$ is a critical point for the loss corresponding to $(x_i, y_i')_{i=1}^n$, but not a critical point for the loss corresponding to $(x_i, y_i)_{i=1}^n$.

Now assume that $n \geq 2 + (d+1)m$. In this case $\ker M'$ is non-trivial, so we can find $v' \in \ker M' \setminus \{0\}$, and then $(y_i')_{i=1}^n$ such that $[\partial_p \ell(H(\theta_{\text{narr}}, x_i), y_i')]_{i=1}^n$ is a non-zero multiple of $v'$. Then $\theta_{\text{wide}}$ is a critical point at which the loss function is non-zero. Thus, by Lemma A.1.2 it is a saddle. $\square$

**Proposition A.2.4** (Proposition 4.2.2 in Section 4.2). *Given samples $(x_i, y_i)_{i=1}^n$ with $x_i \neq 0$ for all $i$ and $x_i \pm x_j \neq 0$ for $1 \leq i < j \leq n$. Given integers $\{m_l\}_{l=1}^L, \{m_l'\}_{l=1}^L$ such that $m_l < m_l'$ for every $1 \leq l \leq L$. Consider two $L$ hidden layer neural networks with input dimension $d$, hidden layer widths $\{m_l\}_{l=1}^L, \{m_l'\}_{l=1}^L$, and output dimension $D$. Denote their parameters by $\theta_{narr}, \theta_{wide}$, respectively. Let $\theta_{narr}$ be a critical point of the loss function corresponding to the samples $(x_i, y_i)_{i=1}^n$, such that $R(\theta_{narr}) \neq 0$. Denote the following sets:*

$$E = \left\{\theta_{wide} = ((a_j')_{j=1}^D, \theta_{wide}^{(L)}) : H(\theta_{wide}, \cdot) = H(\theta_{narr}, \cdot), a_j' = (a_{j1}, ..., a_{jm_L}, 0, ..., 0)\right\};$$

$$E^* = \{\theta_{wide} \in E : \nabla R(\theta_{wide}) = 0\}.$$

*Namely, $E$ is a set of parameters preserving output function, $E^*$ is the set of parameters in $E$ also preserving criticality. Then $E^* \neq E$. Furthermore, $E^*$ contains saddles.*

492   *Proof.* We first show by induction that there is a parameter $\theta_{\text{wide}}^{(L-1)}$ such that

$$H^{(L-1)}(\theta_{\text{wide}}^{(L-1)}, x_i) \neq 0 \qquad\qquad \forall\, 1 \leq i \leq n,$$
$$H^{(L-1)}(\theta_{\text{wide}}^{(L-1)}, x_i) \pm H^{(L-1)}(\theta_{\text{wide}}^{(L-1)}, x_j) \neq 0 \qquad\qquad \forall\, 1 \leq i < j \leq n.$$

493   According to our notation for neural networks (Section 3.1), we denote the entries of $\theta_{\text{narr}}$ as

$$\theta_{\text{narr}} = \left( (a_{jk})_{j,k_L=1}^{D,m_L}, (w_{k_L}^{(L)})_{k_L=1}^{m_L}, ..., (w_{k_1}^{(1)})_{k_1=1}^{m_1}, \theta^{(0)} \right).$$

494   Start with $l = 1$. The linear independence of neurons (Lemma A.1.1) guarantees the existence of
495   some $w_{m_1+1}'^{(1)}, ..., w_{m_1'}'^{(1)}$ such that for every $m_1 < k_1 \leq m_1'$, we have $\sigma(w_{k_1}'^{(1)} \cdot x_i) \pm \sigma(w_{k_1}'^{(1)} \cdot x_j) \neq 0$
496   for $1 \leq i < j \leq n$. Define

$$\theta_{\text{wide}}^{(1)} =: \left( w_{k_1}'^{(1)} \right)_{k_1=1}^{m_1'} = \left( w_1^{(1)}, ..., w_{m_1}^{(1)}, w_{m_1+1}'^{(1)}, ..., w_{m_1'}'^{(1)} \right).$$

497   Then the first layer neuron $H^{(1)}(\theta_{\text{wide}}^{(1)}, x) = [\sigma(w_{k_1} \cdot x)]_{k_1=1}^{m_1'}$ satisfies (a) $H_{k_1}^{(1)}(\theta_{\text{wide}}^{(1)}, \cdot) = $
498   $H_{k_1}^{(1)}(\theta_{\text{narr}}, \cdot)$ for $1 \leq k_1 \leq m_1$, (b) $H^{(1)}(\theta_{\text{wide}}^{(1)}, x_i) \neq 0$ for all $1 \leq i \leq n$ and (c)
499   $H^{(1)}(\theta_{\text{wide}}^{(1)}, x_i) \pm H^{(1)}(\theta_{\text{wide}}^{(1)}, x_i) \neq 0$ for $1 \leq i < j \leq n$. Assume that for some $l \in \{1, ..., L-1\}$
500   we have found $\theta_{\text{wide}}^{(l)}$ such that the following holds:

501      (a) $H_{k_l}^{(l)}(\theta_{\text{wide}}^{(l)}, x) = H_{k_l}^{(l)}(\theta_{\text{narr}}, x)$ for $1 \leq k_l \leq m_l$.

502      (b) $H^{(l)}(\theta_{\text{wide}}^{(l)}, x_i) \neq 0$ for all $1 \leq i \leq n$.

503      (c) $H^{(l)}(\theta_{\text{wide}}^{(l)}, x_i) \pm H^{(l)}(\theta_{\text{wide}}^{(l)}, x_j) \neq 0$ for $1 \leq i < j \leq n$.

504   Then, for the construction of $\theta_{\text{wide}}^{(l+1)}$ we do the following:

505      • For each $1 \leq k_{l+1} \leq m_{l+1}$, set $w_{k_{l+1}}'^{(l+1)} = (w_{k_{l+1}}^{(l+1)}, 0)$.

506      • For each $m_{l+1} < k_{l+1} \leq m_{l+1}'$, find $w_{k_{l+1}}'^{(l+1)} \in \mathbb{R}^{m_l'}$ such that $\sigma\left( w_{k_{l+1}}'^{(l+1)} H^{(l)}(\theta_{\text{wide}}^{(l)}, x_i) \right) \neq$
507       $0$ for all $i$ and $\sigma\left( w_{k_{l+1}}'^{(l+1)} H^{(l)}(\theta_{\text{wide}}^{(l)}, x_i) \right) \pm \sigma\left( w_{k_{l+1}}'^{(l+1)} H^{(l)}(\theta_{\text{wide}}^{(l)}, x_j) \right) \neq 0$ for $1 \leq i <$
508       $j \leq n$. The existence of $w_{k_{l+1}'}^{(l+1)}$ is due to the linear independence of the neurons
509       $\left\{ w \mapsto \sigma\left( wH^{(l)}(\theta_{\text{wide}}^{(l)}, x_i) \right) \right\}_{i=1}^{n}$ from our induction hypothesis (b).

510   Set $\theta_{\text{wide}}^{(l+1)} = \left( (w_{k_{l+1}}'^{(l+1)})_{k_{l+1}=1}^{m_{l+1}'}, \theta_{\text{wide}}^{(l)} \right)$. We have

$$\sigma\left( w_{k_{l+1}}^{(l+1)'} \cdot H^{(l)}(\theta_{\text{wide}}^{(l)}, x) \right) = \sigma\left( \sum_{k_l=1}^{m_l} w_{k_{l+1}k_l}^{(l+1)} \cdot H_{k_l}^{(l)}(\theta_{\text{narr}}, x) + 0 H_{m_l'}^{(l)}(\theta_{\text{wide}}^{(l)}, x) \right)$$
$$= \sigma\left( w_{k_{l+1}}^{(l+1)} \cdot H^{(l)}(\theta_{\text{narr}}, x) \right), \quad \forall\, 1 \leq k_{l+1} \leq m_{l+1},$$
$$H^{(l+1)}(\theta_{\text{wide}}^{(l+1)}, x_i) \pm H^{(l+1)}(\theta_{\text{wide}}^{(l+1)}, x_j) \neq 0, \qquad\qquad \forall\, 1 \leq i < j \leq n$$

511   Namely, (a), (b) and (c) are satisfied for $H^{(l+1)}(\theta_{\text{wide}}^{(l+1)}, x)$, thus proving the induction step.

512   Recall that the (wider) neural network takes the form

$$H(\theta_{\text{wide}}, x) = [H_j(\theta_{\text{wide}}, x)]_{j=1}^{D} = \left[ \sum_{k=1}^{m_L'} a_{jk} H^{(L)}(\theta_{\text{wide}}^{(L)}, x) \right]_{j=1}^{D}.$$

For any $\theta_{\text{wide}}^{(L-1)}$ such that $H_{k_{L-1}}^{(L-1)}\left(\theta_{\text{wide}}^{(L-1)}, x\right) = H_{k_{L-1}}^{(L-1)}\left(\theta_{\text{narr}}^{(L-1)}, x\right)$ for all $1 \leq k_{L-1} \leq m_{L-1}$,

define $E(\theta_{\text{wide}}^{(L-1)})$ as the set of parameters $\theta_{\text{wide}} = ((a'_j)_{j=1}^{D}, (w_{k_L}'^{(L)})_{k_L=1}^{m'_L}, \theta_{\text{wide}}^{(L-1)})$ with the following properties:

- For each $1 \leq j \leq D$, $a'_j = (a_{j1}, ..., a_{jm_L}, 0, ..., 0)$.

- For each $1 \leq k_L \leq m_L$, $w_{k_L}'^{(L)} = (w_{k_L}^{(L)}, 0)$.

- For each $m_L < k_L \leq m'_L$, $w_{k_L}'^{(L)} \in \mathbb{R}^{m'_{L-1}}$ is arbitrary.

Then define
$$E^*(\theta_{\text{wide}}^{(L-1)}) = \left\{ \theta_{\text{wide}} \in E(\theta_{\text{wide}}^{(L-1)}) : \nabla R(\theta_{\text{wide}}) = 0 \right\}.$$

Clearly, $E(\theta_{\text{wide}}^{(L-1)})$ is a connected subset of $E$ of dimension $\geq 1$ and $E^*(\theta_{\text{wide}}^{(L-1)})$ is a subset of $E^*$. We would like to show that for some $\theta_{\text{wide}}^{(L-1)}$, $\nabla R$ is not constant zero on $E(\theta_{\text{wide}}^{(L-1)})$. This means the restriction of $\nabla R$ to $E(\theta_{\text{wide}}^{(L-1)})$ is not constant zero, whence has zero measure in $E(\theta_{\text{wide}}^{(L-1)})$. Let $\theta_{\text{wide}}^{(L-1)}$ be constructed as above. Fix $\theta_{\text{wide}} \in E(\theta_{\text{wide}}^{(L-1)})$. For each $\bar{j}$ consider the partial derivative of the loss function against $a_{\bar{j}m'_L}$:

$$\frac{\partial R}{\partial a_{\bar{j}m'_L}}(\theta_{\text{wide}}) = 2\sum_{i=1}^{n} e_{i\bar{j}}\sigma\left(w_{m'_L}'^{(L)} \cdot H^{(L-1)}(\theta_{\text{wide}}^{(L-1)}, x_i)\right),$$

where
$$e_{i\bar{j}} = \partial_{\bar{j}}\ell\left(H(\theta_{\text{wide}}, x_i), y_i\right) = \partial_{\bar{j}}\ell\left(H(\theta_{\text{narr}}, x_i), y_i\right), \quad \forall 1 \leq i \leq n.$$

The second equality holds because by definition the parameters in $E$ preserve output function. Similar to the proof for Proposition A.2.2, we define an analytic function

$$\varphi(w) = \sum_{i=1}^{n} e_{i\bar{j}}\sigma\left(w \cdot H^{(L-1)}(\theta_{\text{wide}}^{(L-1)}, x_i)\right), \quad w \in \mathbb{R}^{m'_{L-1}}.$$

Note that $\frac{\partial R}{\partial a_{\bar{j}m'_L}}(\theta_{\text{wide}}) = 0$ if and only if $w_{m'_L}'^{(L)} \in \varphi^{-1}(0)$. Since $R(\theta_{\text{narr}}) \neq 0$, there must be some $i$ with $e_{i\bar{j}} \neq 0$. Since $H^{(L-1)}(\theta_{\text{wide}}^{(L-1)}, x_i) \neq 0$ for all $i$ and $H^{(L-1)}(\theta_{\text{wide}}^{(L-1)}, x_i) \pm H^{(L-1)}(\theta_{\text{wide}}^{(L-1)}, x_j) \neq 0$ for $1 \leq i < j \leq n$, the functions

$$\left\{ w \mapsto \sigma\left(w \cdot H^{(L-1)}(\theta_{\text{wide}}^{(L-1)}, x_i)\right) \right\}$$

are linearly independent. Therefore, $\varphi$ is a non-trivial linear combination of analytic, linearly independent functions, so it is analytic and not constant zero. This means $\varphi^{-1}(0)$ has zero measure in $\mathbb{R}^d$. In particular, $\frac{\partial R}{\partial a_{\bar{j}m_L}}$ is not constant zero on $E(\theta_{\text{wide}}^{(L-1)})$, so neither is the restriction of $\nabla R$ to $E(\theta_{\text{wide}}^{(L-1)})$, proving our claim.

Our proof above shows that for any $\theta_{\text{wide}} \in E^*(\theta_{\text{wide}}^{(L-1)})$ and any neighborhood $U$ of $\theta_{\text{wide}}$ we have $U \cap \left(E(\theta_{\text{wide}}^{(L-1)}) \setminus E^*(\theta_{\text{wide}}^{(L-1)})\right) \neq \emptyset$. Meanwhile, the loss function is constant on $E(\theta_{\text{wide}}^{(L-1)})$. Thus, by Lemma A.1.2 we conclude that $\theta_{\text{wide}}$ is a saddle. $\qquad\square$

**Lemma A.2.1.** *Given $\theta_{narr}$. Let $\theta_{wide}^{(L-1)}$ be constructed as in Proposition A.2.4. Let $\theta_{wide} \in E(\theta_{wide}^{(L-1)})$. Then for any $j \in \{1, ..., D\}$ and $k_L \in \{1, ..., m_L\}$ we have $\frac{\partial H}{\partial a'_{jk_L}}(\theta_{wide}, \cdot) = \frac{\partial H}{\partial a_{jk_L}}(\theta_{narr}, \cdot)$. Moreover, for any $l \in \{1, ..., L\}$ the following holds:*

- *For each $k_l \in \{1, ..., m_l\}$ and $k_{l-1} \in \{1, ..., m_{l-1}\}$ we have $\frac{\partial H}{\partial w_{k_l k_{l-1}}'^{(l)}}(\theta_{wide}, \cdot) = \frac{\partial H}{\partial w_{k_L k_{l-1}}^{(l)}}(\theta_{narr}, \cdot)$.*

543     • *For each $k_l > m_l$ we have $\frac{\partial H}{\partial w'^{(l)}_{k_l k_{l-1}}}(\theta_{\text{wide}}, \cdot) = 0$.*

544     *Proof.* The proof is basically straightforward computations. By definition we have

$$\frac{\partial H}{\partial a'_{jk_L}}(\theta_{\text{wide}}, x) = \sigma\left(w'^{(L)}_{k_L} \cdot H^{(L-1)}(\theta^{(L-1)}_{\text{wide}}, x)\right). \tag{4}$$

545     Recall that in our construction, $w'^{(L)}_{k_L} = (w^{(L)}_{k_L}, 0)$ and $H^{(L-1)}_{k_{L-1}}(\theta^{(L-1)}_{\text{wide}}, x) = H^{(L-1)}_{k_{L-1}}(\theta^{(L-1)}_{\text{narr}}, x)$ for
546     all $1 \le k_{L-1} \le m_{L-1}$, whence

$$\frac{\partial H}{\partial a'_{jk_L}}(\theta_{\text{wide}}, x) = \sigma\left(\sum_{k_{L-1}=1}^{m_{L-1}} w^{(L)}_{k_L k_{L-1}} H^{(L-1)}_{k_{L-1}}(\theta^{(L-1)}_{\text{narr}}, x)\right) = \frac{\partial H}{\partial a_{jk_L}}(\theta_{\text{narr}}, \cdot).$$

547     This proves the first part of the lemma.

548     To prove the result for $\frac{\partial H}{\partial w'^{(l)}_{k_l k_{l-1}}}(\theta_{\text{wide}}, \cdot)$ we observe that

$$\frac{\partial H}{\partial w'^{(l)}_{k_l k_{l-1}}}(\theta_{\text{wide}}, x) = A' D'^{(L)} W'^{(L)} ... D'^{(l+1)} \begin{pmatrix} w'^{(l+1)}_{1k_l} \\ \vdots \\ w'^{(l+1)}_{m'_{l+1}k_l} \end{pmatrix}$$
$$\cdot \sigma'\left(w'^{(l)}_{k_l} \cdot H^{(l-1)}(\theta^{(l-1)}_{\text{wide}}, x)\right) H^{(l-1)}_{k_{l-1}}(\theta^{(l-1)}_{\text{wide}}, x)$$

$$\frac{\partial H}{\partial w^{(l)}_{k_l k_{l-1}}}(\theta_{\text{narr}}, x) = A D^{(L)} W^{(L)} ... D^{(l+1)} \begin{pmatrix} w^{(l+1)}_{1k_l} \\ \vdots \\ w^{(l+1)}_{m'_{l+1}k_l} \end{pmatrix}$$
$$\cdot \sigma'\left(w^{(l)}_{k_l} \cdot H^{(l-1)}(\theta^{(l-1)}_{\text{narr}}, x)\right) H^{(l-1)}_{k_{l-1}}(\theta^{(l-1)}_{\text{narr}}, x).$$

549     where $A'$, $A$ are the matrices whose rows are $a'_j$, $a_j$'s:

$$A' = \begin{pmatrix} - & a'_1 & - \\ & \vdots & \\ - & a'_D & - \end{pmatrix}, \quad A = \begin{pmatrix} - & a_1 & - \\ & \vdots & \\ - & a_D & - \end{pmatrix}$$

550     and for each $1 \le \bar{l} \le L$ we define

$$D'^{(\bar{l})} = \begin{pmatrix} \sigma'\left(w'^{(\bar{l})}_1 \cdot H^{(\bar{l}-1)}(\theta^{(\bar{l}-1)}_{\text{wide}}, x)\right) & & \\ & \ddots & \\ & & \sigma'\left(w'^{(\bar{l})}_{m_{\bar{l}}} \cdot H^{(\bar{l}-1)}(\theta^{(\bar{l}-1)}_{\text{wide}}, x)\right) \end{pmatrix},$$

$$D^{(\bar{l})} = \begin{pmatrix} \sigma'\left(w^{(\bar{l})}_1 \cdot H^{(\bar{l}-1)}(\theta^{(\bar{l}-1)}_{\text{narr}}, x)\right) & & \\ & \ddots & \\ & & \sigma'\left(w^{(\bar{l})}_{m_{\bar{l}}} \cdot H^{(\bar{l}-1)}(\theta^{(\bar{l}-1)}_{\text{narr}}, x)\right) \end{pmatrix},$$

$$W'^{(\bar{l})} = \begin{pmatrix} - & w'^{(\bar{l})}_1 & - \\ & \vdots & \\ - & w'^{(\bar{l})}_{m_{\bar{l}}} & - \end{pmatrix},$$

$$W^{(\bar{l})} = \begin{pmatrix} - & w^{(\bar{l})}_1 & - \\ & \vdots & \\ - & w^{(\bar{l})}_{m_{\bar{l}}} & - \end{pmatrix}.$$

551     Again, recall that $w'^{(l+1)}_{k_{l+1}} = (w^{(l+1)}_{k_{l+1}}, 0)$. In particular, when $k_l > m_l$ we have $w'^{(l+1)}_{k_{l+1}k_l} = 0$. Thus,

$$\sigma'\left(w'^{(l)}_{k_l} \cdot H^{(l-1)}(\theta^{(l-1)}_{\text{wide}}, x)\right) H^{(l-1)}_{k_{l-1}}(\theta^{(l-1)}_{\text{wide}}, x) \begin{pmatrix} w'^{(l+1)}_{1k_l} \\ \vdots \\ w'^{(l+1)}_{m'_{l+1}k_l} \end{pmatrix} = 0 \in \mathbb{R}^{m'_{l+1}},$$

552     which shows $\frac{\partial H}{\partial w'^{(l)}_{k_l k_{l-1}}}(\theta_{\text{wide}}, x) = 0$ when $k_l > m_l$. Now let $k_l \le m_l$ and $k_{l-1} \in \{1, ..., m_{l-1}\}$. For

553     each $l < \bar{l} \le L$ define

$$v'^{(\bar{l})} = W'^{(\bar{l})} D'^{(\bar{l})} ... W'^{(l+1)} D'^{(l+1)} \begin{pmatrix} w'^{(l+1)}_{1k_l} \\ \vdots \\ w'^{(l+1)}_{m'_{l+1}k_l} \end{pmatrix}$$
$$\cdot \sigma'\left(w'^{(l)}_{k_l} \cdot H^{(l-1)}(\theta^{(l-1)}_{\text{wide}}, x)\right) H^{(l-1)}_{k_{l-1}}(\theta^{(l-1)}_{\text{wide}}, x)$$

$$v^{(\bar{l})} = W^{(\bar{l})} D^{(\bar{l})} ... W^{(l+1)} D^{(l+1)} \begin{pmatrix} w^{(l+1)}_{1k_l} \\ \vdots \\ w^{(l+1)}_{m'_{l+1}k_l} \end{pmatrix}$$
$$\cdot \sigma'\left(w^{(l)}_{k_l} \cdot H^{(l-1)}(\theta^{(l-1)}_{\text{narr}}, x)\right) H^{(l-1)}_{k_{l-1}}(\theta^{(l-1)}_{\text{narr}}, x),$$

554     anbd similarly, define

$$v'^{(l)} = \sigma'\left(w'^{(l)}_{k_l} \cdot H^{(l-1)}(\theta^{(l-1)}_{\text{wide}}, x)\right) H^{(l-1)}_{k_{l-1}}(\theta^{(l-1)}_{\text{wide}}, x) \begin{pmatrix} w'^{(l+1)}_{1k_l} \\ \vdots \\ w'^{(l+1)}_{m'_{l+1}k_l} \end{pmatrix},$$

$$v^{(l)} = \sigma'\left(w^{(l)}_{k_l} \cdot H^{(l-1)}(\theta^{(l-1)}_{\text{narr}}, x)\right) H^{(l-1)}_{k_{l-1}}(\theta^{(l-1)}_{\text{narr}}, x) \begin{pmatrix} w^{(l+1)}_{1k_l} \\ \vdots \\ w^{(l+1)}_{m'_{l+1}k_l} \end{pmatrix}$$

555     We shall first prove that the first $m_{\bar{l}}$ entries of $v'^{(\bar{l})}$ and the first $m_{\bar{l}}$ entries of $v^{(\bar{l})}$ coincide for each

556     $l \le \bar{l} \le L$. The key is that by our construction of $\theta^{(L-1)}_{\text{wide}}$, for any $1 \le \bar{l} \le L$ and any $k_{\bar{l}} \le m_{\bar{l}}$ we

557     have

$$\sigma'\left(w'^{(\bar{l})}_{k_{\bar{l}}} \cdot H^{(\bar{l}-1)}(\theta^{(\bar{l}-1)}_{\text{wide}}, x)\right) = \sigma'\left(w^{(\bar{l})}_{k_{\bar{l}}} \cdot H^{(\bar{l}-1)}(\theta^{(\bar{l}-1)}_{\text{narr}}, x)\right).$$

558     Since we also have $H^{(l-1)}_{k_{l-1}}(\theta^{(l-1)}_{\text{wide}}, x) = H^{(l-1)}_{k_{l-1}}(\theta^{(l-1)}_{\text{narr}}, x)$ and $w'^{(l)}_{k_{l+1}k_l} = w^{(l)}_{k_{l+1}k_l}$ for $1 \le k_{l+1} \le$

559     $m_{l+1}$, our claim clearly holds for $v'^{(l)}$ and $v^{(l)}$. Suppose the result holds for some $\bar{l} < L$. Then we

560     can write $v'^{(\bar{l})}$ as $v'^{(\bar{l})} = (v^{(\bar{l})}, u)^{\mathrm{T}}$ for some vector $u$. Then

$$v'^{(\bar{l}+1)} = W'^{(\bar{l}+1)} D'^{(\bar{l}+1)} v'^{(\bar{l})}$$

$$= W'^{(\bar{l}+1)} \begin{pmatrix} D^{(\bar{l}+1)} v^{(\bar{l})} \\ \mathrm{diag}\left[\sigma'\left(w'^{(l\mp1)}_{m_{\bar{l}}+1} \cdot H^{(\bar{l}+1)}(\theta^{(\bar{l}+1)}_{\mathrm{wide}}, x)\right)\right]_{k_{\bar{l}+1} > m_{\bar{l}}} u \end{pmatrix}$$

$$= \begin{pmatrix} W^{(\bar{l}+1)} D^{(\bar{l}+1)} v^{\bar{l}} \\ \begin{pmatrix} - & w'^{(\bar{l}+1)}_{m_{\bar{l}+1}+1} & - \\ & \vdots & \\ - & w'^{(\bar{l}+1)}_{m'_{\bar{l}+1}} & - \end{pmatrix} \mathrm{diag}\left[\sigma'\left(w'^{(l\mp1)}_{m_{\bar{l}}+1} \cdot H^{(\bar{l}+1)}(\theta^{(\bar{l}+1)}_{\mathrm{wide}}, x)\right)\right]_{k_{\bar{l}+1} > m_{\bar{l}}} u \end{pmatrix}$$

$$= \begin{pmatrix} v^{(\bar{l}+1)} \\ \begin{pmatrix} - & w'^{(\bar{l}+1)}_{m_{\bar{l}+1}+1} & - \\ & \vdots & \\ - & w'^{(\bar{l}+1)}_{m'_{\bar{l}+1}} & - \end{pmatrix} \mathrm{diag}\left[\sigma'\left(w'^{(l\mp1)}_{m_{\bar{l}}+1} \cdot H^{(\bar{l}+1)}(\theta^{(\bar{l}+1)}_{\mathrm{wide}}, x)\right)\right]_{k_{\bar{l}+1} > m_{\bar{l}}} u \end{pmatrix}.$$

561     This completes the induction step. Finally,

$$\frac{\partial H}{\partial w'^{(l)}_{k_l k_{l-1}}}(\theta_{\mathrm{wide}}, x) = A' v'^{(L)} = \left[A, O_{D \times (m'_L - m_L)}\right] v'^{(L)}$$

$$= A v^{(L)} = \frac{\partial H}{\partial w^{(l)}_{k_l k_{l-1}}}(\theta_{\mathrm{narr}}, x),$$

562     completing the proof. $\qquad\square$

**Proposition A.2.5** (Theorem 4.2.2 in Section 4.2). *Assume that $\ell : \mathbb{R}^2 \to \mathbb{R}$ satisfies: the range of $\partial_p \ell(p, \cdot)$ contains a neighborhood around $0 \in \mathbb{R}^D$. Given $\theta_{narr}$. Let $\theta^{(L-1)}_{wide}$ be constructed as in Proposition A.2.4. Let $N$ denote the parameter size of the narrower network.*

    *(a) Consider sample size $n \geq \frac{1+N}{D}$. For any fixed $(x_i)_{i=1}^n \in \mathbb{R}^{nd}$ with $x_i \pm x_j \neq 0$ and for a.e. $\theta_{wide} \in E(\theta^{(L-1)}_{wide})$, there are sample outputs $(y_i)_{i=1}^n, (y'_i)_{i=1}^n$ such that $\theta_{wide}$ is a critical point for the loss function corresponding to $(x_i, y'_i)_{i=1}^n$ but not so to $(x_i, y_i)_{i=1}^n$.*

    *(b) Consider sample size $n \geq \frac{1+D+\sum_{l=2}^L m_l(m'_{l-1} - m_{l-1}) + N}{D}$. Then we can choose $(y'_i)_{i=1}^n$ so that $E(\theta^{(L-1)}_{wide})$ contains saddles.*

571     *Proof.* The proof is almost identical to that of Proposition A.2.2.

572     (a) Define $M$ as an $N$-rows, $Dn$-columns block matrix

$$M = [D_\theta H(\theta_{\mathrm{narr}}, x_1) \dots D_\theta H(\theta_{\mathrm{narr}}, x_n)].$$

573     For any samples $S =: (x_i, y_i)_{i=1}^n$ we have $\nabla R_S(\theta_{\mathrm{narr}}) = 0$ if and only if

$$M \begin{pmatrix} \partial_p \ell(H(\theta_{\mathrm{narr}}, x_1), y_1) \\ \vdots \\ \partial_p \ell(H(\theta_{\mathrm{narr}}, x_n), y_n) \end{pmatrix} = 0 \in \mathbb{R}^N,$$

574     where $\partial_p \ell$ denotes the gradient of $\ell$ with respect to its first entry. Since $n \geq \frac{1+N}{D}$, $M$
575     has more columns than rows and $\ker M$ is non-trivial. Fix any $v \in \ker M \setminus \{0\}$ and find
576     $(y_i)_{i=1}^n$ such that the (vectorized) vector of partial derivatives $[\partial_p \ell(H(\theta_{\mathrm{wide}}, x_i), y_i)]_{i=1}^n$ is
577     a non-zero multiple of $v$. Thus, $\partial_j \ell(H(\theta_{\mathrm{narr}}, x_i), y_i) \neq 0$ for some $i, j$. Recall that our

construction of $\theta_{\text{wide}}^{(L-1)}$ implies $H^{(L-1)}(\theta_{\text{wide}}^{(L-1)}, x_i) \pm H^{(L-1)}(\theta_{\text{wide}}^{(L-1)}, x_j) \neq 0$. By Lemma A.1.1, the analytic function

$$\varphi : w \mapsto \sum_{i=1}^{n} \partial_j \ell(H(\theta_{\text{wide}}, x_i), y_i) \sigma \left( w \cdot H^{(L-1)}(\theta_{\text{wide}}^{(L-1)}, x_i) \right)$$

is not constant zero. Thus, for a.e. $w' \in \mathbb{R}^{m'_L}$ we have $\varphi(w') \neq 0$. In particular, the set

$$\left\{ \theta_{\text{wide}} \in E(\theta_{\text{wide}}^{(L-1)}) : w'^{(L)}_{m'_L} \notin \varphi^{-1}(0) \right\}$$

has full-measure in $E(\theta_{\text{wide}}^{(L)})$. Note that any $\theta_{\text{wide}}$ in this set is not a critical point of the loss function corresponding to $(x_i, y_i)_{i=1}^{n}$, because the partial derivative for $a'_{jm'_L}$ is non-zero (see also (4) for the formula of $\frac{\partial H}{\partial a'_{jk_L}}$).

Fix $\theta_{\text{wide}}$ in this set. Define

$$M' = [D_\theta H(\theta_{\text{wide}}, x_1) \dots D_\theta H(\theta_{\text{wide}}, x_n)].$$

By Lemma A.2.1, part of each submatrix $D_\theta H(\theta_{\text{wide}}, x_i)$ of $M'$ is $D_\theta H(\theta_{\text{narr}}, x_i)$. In particular, by rearranging the rows if necessary $M'$ can be written as the following block matrix

$$M' = \begin{pmatrix} M \\ U \end{pmatrix}.$$

Let $v' \in \ker M'$ and find some $(y'_i)_{i=1}^{n}$ such that $[\partial_p \ell(H(\theta_{\text{wide}}, x_i), y_i)]_{i=1}^{n}$ is a non-zero multiple of $v'$. Then

$$M' \begin{pmatrix} \partial_p \ell(H(\theta_{\text{narr}}, x_1), y'_1) \\ \vdots \\ \partial_p \ell(H(\theta_{\text{narr}}, x_n), y'_n) \end{pmatrix} = 0,$$

which implies that $\theta_{\text{wide}}$ is a critical point of the loss corresponding to $(x_i, y'_i)_{i=1}^{n}$.

(b) By Lemma A.2.1, the entries of $U$ consists of the following:

  i) $\frac{\partial H}{\partial w'^{(l)}_{k_l k_{l-1}}}(\theta_{\text{wide}}, x_i)$ for $k_l < m_l$, $k_{l-1} > m_{l-1}$ and $1 \leq i \leq n$.

  ii) $\frac{\partial H}{\partial a'_{jk_L}}(\theta_{\text{wide}}, x_i)$ for $k_L > m_L$ and $1 \leq i \leq n$.

The first part gives $\sum_{l=2}^{L} m_l(m'_{l-1} - m_{l-1})$ number of rows of $U$, while the second part gives $D(m'_{l-1} - m_l)$ number of rows of $U$. However, for any $\theta_{\text{wide}} \in E(\theta_{\text{wide}}^{(L-1)})$ such that $w'^{(L)}_{m_L+1} = \dots = w'^{(L)}_{m'_L}$, this reduces to only $D$ different rows (see also (4) for the formula of $\frac{\partial H}{\partial a'_{jk_L}}$). In other words, for such $\theta_{\text{wide}}$ we have a $D + \sum_{l=2}^{L} m_l(m'_{l-1} - m_{l-1}) + N$ row matrix $M''$ with $\ker M'' = \ker M'$. Since $n \geq \frac{1+D+\sum_{l=2}^{L} m_l(m'_{l-1}-m_{l-1})+N}{D}$, $M'$ and $M''$ have more rows than columns, so there is some $v' \in \ker M'' \setminus \{0\}$. Find $(y'_i)_{i=1}^{n}$ such that $[\partial_p \ell(H(\theta_{\text{wide}}, x_i), y_i)]_{i=1}^{n}$ is a non-zero multiple of $v'$. Then

$$M' \begin{pmatrix} \partial_p \ell(H(\theta_{\text{narr}}, x_1), y'_1) \\ \vdots \\ \partial_p \ell(H(\theta_{\text{narr}}, x_n), y'_n) \end{pmatrix} = 0,$$

which implies that $\theta_{\text{wide}}$ is a critical point of the loss corresponding to $(x_i, y'_i)_{i=1}^{n}$. Meanwhile, since $[\partial_p \ell(H(\theta_{\text{wide}}, x_i), y_i)]_{i=1}^{n} \neq 0$, by Assumption 3.2 the loss function is non-zero at $\theta_{\text{wide}}$ (and thus non-zero at $\theta_{\text{narr}}$). It follows from Lemma A.1.2 that $\theta_{\text{wide}}$ is a saddle.

$\square$

