# OpenReview forum: "Uncovering Critical Sets of Deep Neural Networks via Sample-Independent Critical Lifting"
_NeurIPS.cc/2025/Conference — Submitted to NeurIPS 2025_

### Official Review · Reviewer_Y3mK · 2025-06-30

**Clarity:** 3
**Significance:** 2
**Originality:** 2
**Rating:** 2
**Confidence:** 3

**Summary:**

This paper studies the critical points for neural networks, especially how a critical point in a narrow network can be mapped to critical points in a wide network while preserving the output function. For this purpose, the authors expand previous works and provide a broader characterization of sample-independent lift critical points, and also identify the sample-dependent critical points that emerge for sufficiently large sample sizes. Simulations on some simple examples are provided to illustrate the hypothesis.

**Questions:**

1. How important is the analytic assumption in Assumption 3.1? For the given analysis, would second-order differentiability be enough?

2. I don't understand Proposition 4.1.1, can you provide a more mathematical statement with a proper definition of critical embedding operators?

3. The main contribution claimed in the paper is the characterization of sample-dependent critical points. Why would sample-dependent lifted critical points be more important than sample-independent ones?

4. Essentially, the condition for the sample-dependent critical points is that the sample size needs to be larger than the dimension. But wouldn't it be trivial that one has enough degree of freedom so that the gradient of samples can span the whole parameter space, and therefore can easily include all of the critical points?

**Ethical Concerns:**

["NO or VERY MINOR ethics concerns only"]

**Final Justification:**

Overall, the paper is technically solid. My main concern is about the importance of the problem itself, and the authors didn't provide enough justification for its significance. I would like to keep my score.

**Limitations:**

Yes

**Paper Formatting Concerns:**

No concern

**Quality:**

3

**Strengths And Weaknesses:**

Strengths: The theoretical work is solid, and it expands the understanding of the lifted critical points.

Weakness: My major concern is the importance of the problem itself. Although it is a meaningful mathematical problem, understanding the critical lifting operator is not directly relevant to any practically important problem of deep learning. Conceptually, we would hope to obtain a smaller network that mostly preserves the capacity of a large network, while the reverse direction is relatively trivial and less useful. That being said, the motivation of this paper is unclear, which limits its contribution and impact. It would be good to comment on at least the general picture of why the studied problem is important.

---

> ### Author Rebuttal · Authors · 2025-07-30
>
> We thank the reviewer for providing a detailed review and giving valuable suggestions on improving our paper. Below we first give a comment on motivation and importance of the paper, then address the reviewer’s questions.
>
> $\textbf{Motivation and importance of paper:}$ Understanding the global convergence and training dynamics of neural networks remains a fundamental challenge. A key obstacle is the prevalence of non-global critical points and manifolds, which hinder efficient training and convergence to global minima. Although recent work has identified high-dimensional critical manifolds—embedded from narrower networks' critical points—the geometry of these sets and their dependence on training data are still poorly characterized. Without this understanding, it is difficult to analyze the distribution of local minima, saddles, and strict saddles, or to estimate escape probabilities, convergence rates, and acceleration strategies near them.
>
> Our work takes a significant step toward characterizing critical sets geometrically in two ways. First, building on existing work, we demonstrate the prevalence of low-complexity critical points lifted from narrower networks, which exhibit favorable generalization properties. The tendency of training dynamics to stagnate at these low-complexity critical points—combined with early stopping—may help networks generalize well regardless of sample noise. Second, we show that saddles exist among sample-dependent lifted critical points, thus establishing a foundation for further studying escape dynamics from these saddles. We particularly emphasize that for one hidden layer networks, all sample-dependent lifted critical points are saddles, thus narrowing the potential presence of local minima to the sample-independent subset.
>
> $\textbf{Importance of the analytic assumption in Assumption 3.1:}$ The assumption that the activation function is analytic is mainly used to 1.establish the linear independence of neurons (Lemma A.1.1) and 2. guarantee that the level sets of the function has zero measure (Lemma A.1.3). These are used in, e.g., Proposition 4.2.1 to show that any critical point $\theta_{\mathrm{wide}}$ of the form (2) is a saddle, and in Theorem 4.2.1 to construct a sample-dependent lifted critical point $\theta_{\mathrm{wide}}$ of the form (2). A twice-continuously differentiable activation satisfying 1. and 2. also works.
>
> $\textbf{Definition of critical embedding operators:}$ In this paper we mentioned three critical embedding operators, namely the null embedding operator, splitting embedding operator and general compatible embedding operator. Intuitively speaking,
>
>  (a) The null embedding operator adds neurons of zero input weight. In [1] the authors define null embedding operator for neural networks with bias terms. For unbiased neural networks, we need $\sigma(0)=0$ and to set the output weight to be zero.
>
> (b) The splitting embedding operator copies one neuron and “splits” the output weight; for example, a parameter (1/6a, w, 1/3a, w, 1/2a, w) is obtained from (a, w) via splitting embedding operator.
>
>  (c) A general compatible embedding operator generalizes the previous two operators by taking into account, e.g., the composition of them and the permutation of indices in different layers.  We will add more formal definitions of these critical embedding operators in the paper.
>
>  [1] Y. Zhang, Y. Li, Z. Zhang, et al., “Embedding Principle: a hierarchical structure of loss landscape of deep neural networks”, arXiv: 2111.15527, 2021.
>
> $\textbf{Why would sample-dependent lifted critical points be more important than sample-independent ones:}$ The main contribution of this paper is $\textbf{three-fold:}$ introducing a critical lifting operator (contribution (a)), discovering sample-independent lifted critical points which do not arise from previously studied embedding operators (contribution (b)), identifying sample-dependent lifted critical points and show that saddles exist among them (contribution (c)). None of them is prioritized: each advances a different understanding of deep learning.
>
> $\textbf{Essentially, the condition for the sample-dependent critical points is that the sample size needs to be larger than the dimension:}$ Yes, it is the basic idea of the condition for sample-dependent lifted critical points to exist. More rigorously, it is related to the rank and kernel of the network’s Jacobin matrix (which follows from linear independence of neurons), as well as the structure of loss function (which yields the existence of samples). Moreover, we give lower bounds on sample size for sample-dependent lifted critical points to exist, further clarifying the interplay between critical points and samples.

---

> > ### Comment · Reviewer_Y3mK · 2025-08-03
> >
> > I would like to thank the authors for their efforts in addressing my questions. However, given the response, I am still not fully convinced that the current theoretical analysis truly provides critical insights towards the understanding of neural network landscapes, as I discussed in my review. At this point, I am not able to suggest acceptance of the current work.

---

> > > ### Author Response · Authors · 2025-08-05
> > >
> > > Thank you for your feedback. As your current rating suggest that our paper is not technically sound. We really appreciate if you can explicitly point out our technical flaws, weak evaluation or inadequate reproducibility.  Regarding the significance of our results, we are also happy to address any remaining problems. Your response would be incredibly valuable for strengthening our work.

---

### Official Review · Reviewer_CKgN · 2025-06-30

**Clarity:** 3
**Significance:** 2
**Originality:** 2
**Rating:** 4
**Confidence:** 4

**Summary:**

The authors investigate critical points of the loss landscape of multi-layer perceptrons without biases. Specifically, they study so-called lifted critical points that are obtained by embedding a network in a wider network. Some embeddings, like neuron duplication are well-known in the field. Neuron duplication lifts critical points to sample-independent critical points, in the sense that the parameters of the wide network are a critical points for any dataset for which the parameters of the narrow network are a critical point. The main contributions in this paper are: 1) a clear definition of sample-dependent and sample-independent lifted critical points, 2) an example of a three-layer network with zero weights in the first two layers that leads to a sample-independent lifted critical point, which demonstrates that the well-known embeddings are not the only sample-independent lifted critical points, 3) proofs that non-zero-loss sample-dependent lifted critical points with zero output weights exist (given enough data), have zero measure in parameter space, and are saddles (in the case of one hidden layer). The theoretical findings are further supported by a well-designed illustrative experiment.

**Questions:**

- In the example starting on line 142, the condition \sigma(0) = 0 is missing (this is fine in the appendix).
- The construction of sample-independent critical lifting is smart, but took me a moment to digest. I think it would be helpful to explain the reasoning and intuition behind the construction.
- Font size in the figures is too small; I could not read the numbers when printed on A4.
- Figure 2: The three panels in Figure 2 are a bit confusing; in particular because the tick labels are so small. Would it be possible to show the contour plot of the loss landscape in one wide panel, with a2 in [-0.1, 0.1] and w2 in [-1.2, 1.2], and maybe another, zoomed-in panel with w2 in the range [-0.2, 0.2], if the critical points at w2 = 0 and w2 = 0.1236 cannot be distinguished in the wider panel? Also, I think it would be nice to mark sample-dependent and sample-independent points differently (I guess (0, 0) is sample-independent).
- Figure 3: Could you extend the w range to [-1.2, 1.2]? I am curious to see, if the zero curve of phi "bends back", or if there is another reason to explain the critical point at w2=1.0258 in Figure 2. Also, could you mark the critical points that we see in Figure 2 (at t = 0)?
- line 325: 1 \leq i < j \leq n (should be n instead of m)
- line 329: d, m \in \mathbb N
- line 330: non-zero (instead of non=zero)
- I Would move Definition A.1 and Remark A.1 to the beginning of the appendix, because Lemma A.1.1 relies already on analytic functions.
- line 450: refer the reader to Lemma A.1.3.
- formula below line 466: what do the vertical lines above and below \nabla_\theta H indicate?

**Ethical Concerns:**

["NO or VERY MINOR ethics concerns only"]

**Final Justification:**

I appreciate the good discussion with the authors. I fell my continues to be justified and I do not change my overall rating.

**Limitations:**

- In the example starting on line 142, the condition \sigma(0) = 0 is missing (this is fine in the appendix).
- The construction of sample-independent critical lifting is smart, but took me a moment to digest. I think it would be helpful to explain the reasoning and intuition behind the construction.
- Font size in the figures is too small; I could not read the numbers when printed on A4.
- Figure 2: The three panels in Figure 2 are a bit confusing; in particular because the tick labels are so small. Would it be possible to show the contour plot of the loss landscape in one wide panel, with a2 in [-0.1, 0.1] and w2 in [-1.2, 1.2], and maybe another, zoomed-in panel with w2 in the range [-0.2, 0.2], if the critical points at w2 = 0 and w2 = 0.1236 cannot be distinguished in the wider panel? Also, I think it would be nice to mark sample-dependent and sample-independent points differently (I guess (0, 0) is sample-independent).
- Figure 3: Could you extend the w range to [-1.2, 1.2]? I am curious to see, if the zero curve of phi "bends back", or if there is another reason to explain the critical point at w2=1.0258 in Figure 2. Also, could you mark the critical points that we see in Figure 2 (at t = 0)?
- line 325: 1 \leq i < j \leq n (should be n instead of m)
- line 329: d, m \in \mathbb N
- line 330: non-zero (instead of non=zero)
- I Would move Definition A.1 and Remark A.1 to the beginning of the appendix, because Lemma A.1.1 relies already on analytic functions.
- line 450: refer the reader to Lemma A.1.3.
- formula below line 466: what do the vertical lines above and below \nabla_\theta H indicate?

**Quality:**

3

**Strengths And Weaknesses:**

## Strengths and Weaknesses

### Quality
The proofs look sound and well written; it would be good to have them double-checked by a mathematician, however. In particular, the appendix is of higher quality than other works I know in this field.

### Clarity
For a specialist, the paper is clear, but it is pretty dense and formal, and may be hard to digest for a broad audience. See specific comments below.

### Significance
The thorough investigation of sample-dependent and sample-independent critical points in bias-free multi-layer perceptrons is interesting to me, but the findings are quite specialized. For theoreticians, the example of a sample-independent lifted critical point with zero weights in the first two layers, and the existence of sample-dependent critical points with zero output weights, may not be surprising, but a nice contribution to our knowledge about critical points in neural network loss landscapes. The discussion of open theoretical problems is also valuable. For machine learning practitioners I do not see (yet) practical implications of these results.

### Originality
The proof techniques look pretty standard, but the results are novel. The distinction between sample-dependent and sample-independent lifting of critical points is original and useful.

---

> ### Author Rebuttal · Authors · 2025-07-30
>
> We thank the reviewer for providing a detailed review and giving valuable suggestions on improving our paper. We will correct the typos in the revision of this paper. Below we briefly discuss how our work is related to a broader field in machine learning:
>
> Understanding the global convergence and training dynamics of neural networks remains a fundamental challenge. A key obstacle is the prevalence of non-global critical points and manifolds, which hinder efficient training and convergence to global minima. Although recent work has identified high-dimensional critical manifolds—embedded from narrower networks' critical points—the geometry of these sets and their dependence on training data are still poorly characterized. Without this understanding, it is difficult to analyze the distribution of local minima, saddles, and strict saddles, or to estimate escape probabilities, convergence rates, and acceleration strategies near them.
>
> Our work takes a significant step toward characterizing critical sets geometrically in two ways. First, building on existing work, we demonstrate the prevalence of low-complexity critical points lifted from narrower networks, which exhibit favorable generalization properties. The tendency of training dynamics to stagnate at these low-complexity critical points—combined with early stopping—may help networks generalize well regardless of sample noise. Second, we show that saddles exist among sample-dependent lifted critical points, thus establishing a foundation for further studying escape dynamics from these saddles. We particularly emphasize that for one hidden layer networks, all sample-dependent lifted critical points are saddles, thus narrowing the potential presence of local minima to the sample-independent subset.

---

> > ### Comment · Reviewer_CKgN · 2025-08-04
> > **response to reply.**
> >
> > Thank you for your reply. I agree that this paper reports interesting progress on our understanding of critical points in neural network loss landscapes, and I would like to reemphasize that I think it is original and solid work with a particularly high-quality appendix. However, I still think that it speaks to a pretty specialized audience in its current form. I will therefore leave my scores unchanged.

---

> > > ### Author Response · Authors · 2025-08-07
> > >
> > > We want to thank you again for your valuable efforts in reviewing the paper!

---

### Official Review · Reviewer_THr3 · 2025-07-06

**Clarity:** 1
**Significance:** 1
**Originality:** 2
**Rating:** 2
**Confidence:** 2

**Summary:**

The paper study the notions of  sample dependent and independent critical lifting operators following the previous work of Zhang et al on critical embeddings. This operator maps a critical parameter of a narrower neural network to a wider network without changing the functional form of the output of the neural net, as well as it keeps the criticality property. If this map from theta_1 to theta_2 holds for the loss of the neural net given any arbitrary dataset, then theta_1 is a sample independent critical set, otherwise it is sample dependent. The authors  show an example of sample independent critical points that are not covered by the affine embeddings of the work of Zhang, examples of existing sample dependent critical points in general, that for one layer networks under some assumption all sample dependent critical points have to be saddle, and finally some example for saddle points for multi layer networks. They further plot some toy loss functions to illustrate the concepts they defined pictorially.

**Questions:**

Can authors clarify what is the importance of these cases they illustrate of sample independent and dependent critical points for the purpose of understanding neural networks optimization, generalization, or representation power?

What is the importance of the authors conjecture on the non-existence of other sample dependent  critical points? Generally why is the feature of being sample dependent or independent for a critical point should be studied?

**Ethical Concerns:**

["NO or VERY MINOR ethics concerns only"]

**Limitations:**

In addition to the fact that non of their analysis is inclusive and seems to be a sparse set of results and examples, the overall goal of this study and why the community should care about these operators in general is not discussed.

**Paper Formatting Concerns:**

The formatting seems consistent.

**Quality:**

1

**Strengths And Weaknesses:**

The theoretical results not novel to me in that:
The construction of sample independent critical lifting operator in line 145 is trivial and not interesting. It only zeros out two layers and claims the previous affine mappings introduced in work of Zhang do not cover them. Furthermore, they don’t characterize all such sample independent critical lifting operators beyond this trivial example.

The study of sampled dependent critical lifting operator is also not inclusive at all, again they just give a counter example of large enough sample size, that there exist a non-empty sample dependent critical lifting operator. They further show that they such points should be a saddle point in a specific case.

Moreover, the writing is very vague without defining the concepts they use, some arguments are contradictory or illogical. Please see below for few examples:

Other issues:
After line 145: why is the index of w and w’ up to k_3? Isn’t k_3 the dimension for a_1 ?


Remark 4.5 for simplicity assume the activation is even or odd. Which result does this assumption hold for exactly?

The argument of Remark 4.5 does not make sense at all, in particular the phrase  ‘On the other hand, if  w’_k \in \{w_k, -w_k\}_{k=1}^m then \theta_{wide} is a sample-independent lifted critical point. Therefore, up to permutation of the entries, a sample-independent lifted critical point from \theta_{narr} takes the form (2),’


The null-embedding, splitting embedding, and compatible embedding from previous work are not even defined here.

Line 153 and 154 very vague: “we cannot avoid the sample-independent lifted critical points which 154 are not produced by these embedding operators”

The notion of ‘saddle’ is not defined.

---

> ### Author Rebuttal · Authors · 2025-07-30
>
> We thank the reviewer for providing a detailed review and giving valuable suggestions on improving our paper. Below we first give a comment on motivation and importance of the paper, then address the reviewer’s questions.
>
> $\textbf{Motivation and importance of paper:}$
> Understanding the global convergence and training dynamics of neural networks remains a fundamental challenge. A key obstacle is the prevalence of non-global critical points and manifolds, which hinder efficient training and convergence to global minima. Although recent work has identified high-dimensional critical manifolds—embedded from narrower networks' critical points—the geometry of these sets and their dependence on training data are still poorly characterized. Without this understanding, it is difficult to analyze the distribution of local minima, saddles, and strict saddles, or to estimate escape probabilities, convergence rates, and acceleration strategies near them.
>
>  Our work takes a significant step toward characterizing critical sets geometrically in two ways. First, building on existing work, we demonstrate the prevalence of low-complexity critical points lifted from narrower networks, which exhibit favorable generalization properties. The tendency of training dynamics to stagnate at these low-complexity critical points—combined with early stopping—may help networks generalize well regardless of sample noise. Second, we show that saddles exist among sample-dependent lifted critical points, thus establishing a foundation for further studying escape dynamics from these saddles. We particularly emphasize that for one hidden layer networks, all sample-dependent lifted critical points are saddles, thus narrowing the potential presence of local minima to the sample-independent subset.
>
> $\textbf{Why is the index of $w$ and $w'$ up to $k_3$:}$ $k_3$ is both for enumerating the entries of $a_1$ and the $w_{k_3}^{(3)}$'s. The dimension of $a_1$ is $m_3$ or $m_3+1$ depending on the narrower or wider network.
>
> $\textbf{For simplicity assume the activation is even or odd. Which result does this assumption hold for exactly:}$ The result ''By linear independence of neurons …then $\theta_{\mathrm{wide}}$ is a sample-independent lifted critical point''.
>
> $\textbf{The argument of Remark 4.5 does not make sense at all:}$ We apologize for the typo. Here it should be “Therefore, up to permutation of the entries, a $\textit{sample-dependent}$ lifted critical point from $\theta_{\mathrm{narr}}$ takes the form (2)”.
>
> $\textbf{Definition of critical embedding operators:}$ In this paper we mentioned three critical embedding operators, namely the null embedding operator, splitting embedding operator and general compatible embedding operator. Intuitively speaking,
>
>  (a) The null embedding operator adds neurons of zero input weight. In [1] the authors define null embedding operator for neural networks with bias terms. For unbiased neural networks, we need $\sigma(0)=0$ and to set the output weight to be zero.
>
> (b) The splitting embedding operator copies one neuron and “splits” the output weight among these copies; for example, a parameter (1/6a, w, 1/3a, w, 1/2a, w) is obtained from (a, w) via splitting embedding operator.
>
>  (c) A general compatible embedding operator generalizes the previous two operators by taking into account, e.g., the composition of them and the permutation of indices in different layers.  We will add more formal definitions of these critical embedding operators in the paper.
>
>  [1] Y. Zhang, Y. Li, Z. Zhang, et al., “Embedding Principle: a hierarchical structure of loss landscape of deep neural networks”, arXiv:2111.15527, 2021.
>
> $\textbf{Line 153 and 154 very vague:}$ Please refer to the intuitive definition of critical embedding operators. Hopefully this can make the statement clearer.
>
> $\textbf{The notion of ‘saddle’ is not defined:}$ A saddle of a real-valued, differentiable function is a critical point which is neither a local minimum nor a local maximum.
>
> $\textbf{The work is not inclusive:}$ Under mild assumptions on activation, we are able to discover all sample-independent lifted critical points for one hidden layer neural networks. For example when $\sigma$ is odd, we have a linear independence of neurons and their derivatives: \{\sigma(w_k x), \sigma(w_k x)x_1, ..., \sigma(w_k x)x_d: 1 <= k <= m\} are linearly independent for any $m \in \mathbb{N}$ and non-zero $w_k$'s such that $w_k \pm w_j \ne 0$ for distinct $k,j$ (here $x_t$ denotes the t-th entry of input $x$). This result implies that a sample-independent lifted critical point \theta' = (a_k', w_k') from \theta = (a_k, w_k) iff.
>
> (a) w_k' \in \{\pm w_k\}_{k=1}^m for any $k$ such that $a_k' \ne 0$.
>
> (b) w_k' \in \{\pm w_k\}_{k=1}^m \cup \{0\} for any $k$ such that $a_k' = 0$.
>
> (c) Given $k' \in \{1, ..., m'\}$, if $w_{k'}' \in \{\pm w_k\}$ for some $k \in \{1, ..., m\}$, then for any $x \in \mathbb{R}^d$ we have $\sum_{j:w_j' = \pm w_{k'}'} a_j' \sigma(w_{k'}^{'\mathrm{T}} x) = \sum_{j: w_j = \pm w_k} a_j \sigma(w_k^{\mathrm{T}} x)$.
>
> In short, the null embedding, splitting embedding and flipping the signs of $w_k$'s produce all sample-independent lifted critical points. However, the case for deeper networks is unclear to us because the linear independence result is much more complicated. Very few work, e.g. [2] discusses this problem,  and even with this result it is hard to explicitly characterize these critical points.
>
> For sample-dependent lifted critical points, we mentioned in Remark 4.5 that it takes the form (2). However, the case for deeper networks is much more complicated. We hope to find a way to characterize them in the future.
>
> [2] L. Zhang, “Linear Independence of Generalized Neurons and Related Functions”, arXiv:2410.03693, 2024.

---

### Official Review · Reviewer_8KJm · 2025-07-13

**Clarity:** 2
**Significance:** 2
**Originality:** 3
**Rating:** 4
**Confidence:** 2

**Summary:**

This paper develops a new theoretical tool to better understand the relationship between critical points across neural networks with different architectures and their dependence on training data. The authors introduce a new sample-independent critical lifting operator that maps parameters from a narrower to a wider network, preserving both output and criticality regardless of data samples. They show that previous embedding operators do not capture all such critical points. Additionally, they identify output-preserving critical sets that, for large sample sizes, generally contain sample-dependent critical points, which are typically saddle points, especially in multi-layer networks.

**Questions:**

- Is it necessary to discover all sample-independent lifted critical points?

- Concerning the Sample dependence of critical points, it is not clear to me why your work complements the previous studies. Could the authors explain this point more clearly?

**Ethical Concerns:**

["NO or VERY MINOR ethics concerns only"]

**Limitations:**

Yes

**Quality:**

3

**Strengths And Weaknesses:**

Strengths:

- This paper introduces a new sample-independent critical lifting operator with a theoretical guarantee that enables a more general analysis of the loss landscape structure. This operator maps parameters from a narrower neural network to a set of parameters in a wider network, while preserving both the output function and the criticality of the point, independently of the training data.

- This paper also provides examples showing that there exist sample-independent critical points that cannot be generated by previously studied embedding operators (such as Embedding Principle), thereby expanding the understanding of the loss landscape in wider neural networks.

- This paper also identifies a class of output-preserving critical sets containing sample-dependent critical points, primarily saddle points.

Weakenesses:

- However, constructing the critical lifting operator is restricted to feedforward multilayer perceptron (MLP) neural networks, and seems hard to extend to highly non-trivial and complex tasks.

- The paper focuses on theory and simple network examples, providing limited experiments on large, complex networks or real-world data.

- It is hard to follow the proofs without instructions.

---

> ### Author Rebuttal · Authors · 2025-07-30
>
> We thank the reviewer for providing a detailed review and giving valuable suggestions on improving our paper. Below we first give a comment on motivation and importance of the paper, then address the reviewer’s questions.
>
> $\textbf{Motivation and importance of paper:}$
> Understanding the global convergence and training dynamics of neural networks remains a fundamental challenge. A key obstacle is the prevalence of non-global critical points and manifolds, which hinder efficient training and convergence to global minima. Although recent work has identified high-dimensional critical manifolds—embedded from narrower networks' critical points—the geometry of these sets and their dependence on training data are still poorly characterized. Without this understanding, it is difficult to analyze the distribution of local minima, saddles, and strict saddles, or to estimate escape probabilities, convergence rates, and acceleration strategies near them.
>
> Our work takes a significant step toward characterizing critical sets geometrically in two ways. First, building on existing work, we demonstrate the prevalence of low-complexity critical points lifted from narrower networks, which exhibit favorable generalization properties. The tendency of training dynamics to stagnate at these low-complexity critical points—combined with early stopping—may help networks generalize well regardless of sample noise. Second, we show that saddles exist among sample-dependent lifted critical points, thus establishing a foundation for further studying escape dynamics from these saddles. We particularly emphasize that for one hidden layer networks, all sample-dependent lifted critical points are saddles, thus narrowing the potential presence of local minima to the sample-independent subset.
>
> $\textbf{Is it necessary to discover all sample-independent lifted critical points:}$ Yes from the motivation of our paper. Currently, under a mild assumption on activation, we are able to discover all sample-independent lifted critical points for one hidden layer neural networks, as they are all produced by critical embedding operators. The case for deeper networks is unclear to us yet.
>
> $\textbf{Concerning the sample dependence of critical points, it is not clear to me why your work complements the previous studies: }$ Several works, such as [1, 2], discover that embedding operators can produce sample-independent lifted critical points. We include this as Proposition 4.1.1 in our paper. However, they do not notice 1. the sample (in)dependence property of critical embedding, and 2. the operators cannot produce all sample-independent lifted critical points for deep neural networks. We address them in our paper. We also discover sample dependent critical points which are not discovered by previous works.
>
>  [1] Y. Zhang, Z. Zhang, T. Luo, et al., “Embedding Principle of Loss Landscape of Deep Neural Networks”, NeurIPS, 2021.
>
>  [2] B. Simsek, F. Ged, A. Jacot, et al., “Geometry of the Loss Landscape in Overparameterized Neural Networks: Symmetries and Invariances”, ICML, 2021.

---

### Note · Authors · 2025-08-13

We appreciate the constructive feedback from the reviewers. Below we summarize our main contributions, outline key enhancements planned for the final version, and clarify possible misunderstandings.

Our work studies the critical set geometry of neural networks, which is an essential step toward understanding the training behavior of such models. The main contributions of this paper are: introducing a critical lifting operator; discovering sample-independent lifted critical points which do not arise from previously studied embedding operators; identifying sample-dependent lifted critical points and show that saddles exist among them. We provide theoretical results and lucid illustration through a concise toy example.

We deeply appreciate the reviewers’ recognition of our paper’s strengths, such as the novelty and rigor of our theoretical contributions and the comprehensiveness of our appendix. This is indeed a highlight of our paper besides its importance in understanding the training of neural networks. We also thank the valuable suggestions from them. If accepted, we will enhance the final version by:

$\textbf{i)}$ including more detailed motivation of our work

$\textbf{ii)}$ adding definitions of the embedding operators for completeness

$\textbf{iii)}$ emphasizing the result on one hidden layer networks — sample-independent lifted critical points are all produced by embedding operators, and sample-dependent ones are saddles — by writing them as theorems. This also makes our analysis more inclusive.

$\textbf{iv)}$ correcting the typos identified by our reviewers

Finally, we would like to clarify that: $\textbf{1.}$ Our paper is mathematically sound. $\textbf{2.}$ We focus on both sample-independent and sample-dependent lifted critical points, not prioritizing any one of them.


We hope these remarks help the AC and reviewers better appreciate our work. We look forward to submitting an enhanced final version. Thanks you!

---

### Decision · Program_Chairs · 2025-09-17

**Decision:**

Reject

**Comment:**

The paper analyzes critical points in bias-free multi-layer perceptrons, focusing on lifting them from narrower to wider networks. It defines sample-dependent and sample-independent cases, presents a three-layer example beyond known embeddings, and proves that certain sample-dependent points with zero output weights exist for large datasets. The findings are supported by illustrative experiments.

This work presents a sound technical contribution to the theory of neural network loss landscapes, supported by careful analysis and a well-prepared appendix. While two reviewers considered it borderline acceptable, others questioned its significance beyond illustrative examples. The rebuttal clarified key definitions and motivation but did not fully address concerns regarding broader impact. Overall, the work advances theoretical understanding, yet would benefit from broader contextualization and clearer exposition to engage a wider audience. The authors are encouraged to incorporate the valuable feedback provided by the reviewers.